

# Shallow cloud variability in Houston, Texas during the ESCAPE and TRACER field experiments

Zackary Mages[1], Pavlos Kollias[1,2,3], Bernat Puigdomènech Treserras[3], Paloma Borque[3], and Mariko Oue[1]

[1]School of Marine and Atmospheric Sciences, Stony Brook University, Stony Brook, NY, USA
[2]Environmental and Climate Sciences Department, Brookhaven National Laboratory, Upton, NY, USA
[3]Department of Atmospheric and Oceanic Sciences, McGill University, Montreal, QC, Canada

*Correspondence to*:
Zackary Mages (zackary.mages@stonybrook.edu)



## Abstract

Shallow convection plays an important role in Earth's climate system by regulating the vertical transport of heat, moisture, and momentum in the lower troposphere. Aerosols, large-scale meteorology, and low-level convergence influence the spatiotemporal variability of shallow convection, and the coastal urban area of Houston, Texas is an ideal laboratory to investigate these complex interactions. Here, geostationary satellite and ground-based radar observations from June to September 2022 during the TRacking Aerosol Convection interactions ExpeRiment (TRACER) and Experiment of Sea Breeze Convection, Aerosols, Precipitation, and Environment (ESCAPE) field campaigns are used to characterize the spatial coverage, vertical extent and precipitation fraction of shallow convective clouds. The fused operational remote sensing datasets over a 250x250 km domain are evaluated against profiling observations. The domain-wide diurnal shallow cloud fractions are used to identify four distinct modes of shallow convection. In all clusters, the domain-wide cloud fractions are consistently higher than the domain-wide precipitation fractions, and shallow cloud fractions are higher over water than they are over land while the shallow precipitation fractions show the opposite behavior. In the two modes with minimal deep cloud activity, shallow cloud frequency is highest over ocean in the early morning, and there is a transition to higher shallow cloud frequency over land by the afternoon in one cluster or to high shallow cloud frequencies everywhere by the afternoon in the other. Lastly, we find regions with higher shallow cloud top heights and a large region along the coastline where shallow clouds are more likely to precipitate.

## Short Summary

Convective clouds are a key component of the climate system. Using remote sensing observations during two field experiments in Houston, Texas, we identify four diurnal patterns of shallow convective clouds. We find areas more frequently experiencing shallow convective clouds, and we find areas where the vertical extent of shallow convective clouds is higher and where they are more likely to precipitate. This provides insight into the complicated environment that forms these clouds in Houston.



## 1 Introduction

Atmospheric convection is a fundamental mechanism for the vertical transport of heat, moisture, and momentum in the troposphere, significantly impacting the large-scale atmospheric circulation and local environment. Convection also modulates cloudiness, which further influences large-scale atmospheric circulations (Hartmann et al., 1984; Su et al., 2014; Sherwood et al., 2014). Convective clouds are also the source of extreme weather, and several studies have shown that as the climate warms in the future, an increase in the occurrence of extreme precipitation and severe weather associated with convection is anticipated (Trapp et al., 2009; Diffenbaugh et al., 2013; Sillman et al., 2013; Seeley and Romps, 2015).

Several factors contribute to the formation, evolution, and dissipation of convective clouds, the main ones being influences by aerosols, large-scale meteorology, and boundary layer organization (Rosenfeld et al., 2006; Fan et al., 2013; Varble, 2018; Lebo, 2018; Wilson and Schreiber, 1986). A coastal urban area like Houston, Texas is an excellent place to observe the complicated interactions between convection, aerosols, large-scale meteorology, and convergence zones (Jensen et al., 2022). The Houston region is warm and humid in the summer and commonly experiences onshore flow and sea breeze-forced convection, which interacts with a range of aerosol conditions associated with the city's urban and industrial emissions. This argument was initially put forward by the Aerosols, Clouds, Precipitation, and Climate (ACPC) work group, a joint initiative of the International Geosphere-Biosphere Programme and the World Climate Research Programme (Quaas et al., 2015). This effort led to the organization of two large field campaign programs in the Houston area during the 2022 summer period. The National Science Foundation (NSF) sponsored a large field experiment entitled "Experiment of Sea Breeze Convection, Aerosols, Precipitation, and Environment" (ESCAPE; Kollias et al., 2024), which took place between 30 May – 30 September 2022. ESCAPE overlapped with the four-month Intensive Observation Period (IOP) of the United States Department of Energy (DOE) Atmospheric Radiation Measurement (ARM)-funded "TRacking Aerosol Convection interactions ExpeRiment" (TRACER) field campaign (Jensen et al., 2019).

The goal of these field campaigns was to collect high spatial and temporal resolution observations of convective clouds in the region under a wide array of complicated environmental conditions (i.e., clean and polluted aerosol regimes, sea breeze events, urban dynamics, etc.). In Houston, a combination of urban emissions from a city of 7 million people, coupled with emissions from extensive petroleum refining and chemical manufacturing, creates unique aerosol conditions that could influence cloud properties and lifecycles (Lebo, 2018). Houston also has frequent sea breeze circulations with two bay breeze circulations as well (Kocen, 2013). Another consideration to be made is how the urban environment itself influences clouds, precipitation, and the sea breeze circulation. Simpson et al. (2008) showed that Chennai, India impacted the local sea breeze circulation's propagation and precipitation totals by increasing the near surface air temperature over the city in the early morning hours. Specifically in Houston, Chen et al. (2011) showed that the land surface could modify the sea and bay breeze circulations as they show soil moisture, sea surface temperature, and the urban landscape modify the stagnation of the circulations. Houston



may also affect clouds and precipitation by changing surface fluxes, surface roughness, and production of aerosols (Rozoff et
al., 2003; Shepherd, 2005; van den Heever and Cotton, 2007). The change in surface roughness from rural to urban areas may
strengthen convective forcing (Changnon et al., 1981; Bornstein and Lin, 2000; Thielen et al., 2000). The urban heat island of
a city provides surface heating that will impact convective instability and inhibition while also inducing circulations near the
surface that will foster cloud growth (Shepherd et al., 2002; Shepherd and Burian, 2003). Long-term radar observations have
been used to show urban impacts on convective initiation, rainfall, lifetime, and other convective properties (Ashley et al.,
2012; Ganeshan et al., 2013; Haberlie et al., 2015; Kingfield et al., 2018). Lastly, Ganeshan et al. (2013) showed that the
interaction between the urban heat island and sea breeze circulations increased the frequency and intensity of positive rainfall
anomalies.

Shallow cumulus clouds are the most frequently observed cloud type in the Houston area compared to deep convective clouds
(Tuftedal et al., 2024). While most of the previous studies focused on deep convection in this area, precipitation from shallow
convection is non-negligible in terms of the total precipitation in the area (Kumar et al., 2013a; Kumar et al., 2013b). Especially
in humid environments, even shallow convective clouds can produce heavy precipitation with comparable radar reflectivities
to deep convection (May and Ballinger, 2007; Oue et al., 2010). Furthermore, shallow convection can modulate the lower
atmosphere by moistening it, which favors the development of deep convection (Sherwood and Wahrlich, 1999; Derbyshire
et al. 2004; Mapes et al., 2006; Holloway and Neelin, 2009; Nuijens et al., 2009; Powell and Houze, 2013). Aerosol-cloud
interactions on shallow convective clouds have been studied since the 1970s. While they have established the impact of
aerosols on droplet number concentrations and sizes (e.g., Twomey, 1974; Lohmann and Feichter, 2005), much less has been
reported in terms of invigoration (depicted as deepening of the shallow cloud layer) or precipitation modulation. Thus, it is not
well understood what environments, including meteorological, aerosol, and geographical conditions, control the early growth
stage of convective clouds.

Here, we focus on the characterization of the spatiotemporal variability of shallow clouds and associated precipitation. *The
main objective is to characterize the control of meteorology and aerosols on the early growth stage of shallow convective
clouds. Three main properties of shallow cloudiness will be evaluated: cloud fraction, cloud top height, and precipitation
fraction.* Over the four-month IOPs of the ESCAPE and TRACER field campaigns, operational satellite and weather radar
datasets are combined to provide a large statistical sample of shallow clouds and associated precipitation over the Houston
area. Daily statistics are used to infer different patterns of shallow and deep clouds, and observations of them over land and
ocean are used to further elucidate their relationships with surface properties and meteorology. Days with deep clouds are
identified and ignored, allowing for the analysis of the spatiotemporal variability of shallow clouds in the domain, their heights,
and their likelihood to precipitate.

**2 Data**



### 2.1 DOE ARM observations


During the TRACER field campaign, the DOE ARM program deployed the first ARM Mobile Facility (AMF1) at LaPorte Municipal Airport, located 35 km east of the Houston downtown area [29º 40' 12" N, -95º 3' 32.4" E]. The AMF1 operates a comprehensive suite of active and passive remote sensors that can provide continuous, high spatiotemporal resolution information about aerosols, clouds, precipitation, and radiation in the atmosphere above and around the AMF1 location. For

example, the Parsivel2 laser disdrometer at the AMF1 measures particle size distribution over the range of 0.06 - 24 mm and classifies precipitation type (drizzle, drizzle with rain, rain, rain and drizzle with snow, snow, snow grains, freezing rain, and hail) using the precipitation rate (Bartholomew, 2020). Precipitation type and rate are used here (Wang and Shi, 2021). In addition, frequent radiosonde launches were performed at the AMF1. The TRACER campaign followed a radiosonde launch schedule of four per day every six hours (approximately 0:00, 6:00, 12:00, and 18:00 UTC), but this schedule varied slightly

on "enhanced sounding days" with forecasted deep convection. On these days, an additional radiosonde was launched at approximately 21:00 UTC to better capture rapid daytime development of the atmospheric thermodynamic structure (Jensen et al., 2019). Vertical profiles of atmospheric state variables like pressure, temperature, relative humidity, wind speed, and wind direction collected by the radiosondes are used here (Burk, 2021).

Value-added products (VAPs), which provide post-processed data from the suite of instruments at AMF1, are also available. The Active Remote Sensing of Clouds (ARSCL; Clothiaux et al., 2001; Kollias et al., 2016) VAP combines profiling mm-wavelength radar and lidar observations to estimate a feature mask that includes the radar Doppler moments and cloud boundary estimates from laser ceilometers and Micropulse Lidars (MPLs). The Ka-band Zenith-pointing Radar (KAZR; Kollias et al., 2020) is the profiling mm-wavelength radar operating at the AMF1 during TRACER. The KAZR radar

reflectivity and cloud base height estimate from ARSCL are used here (Johnson et al., 2021). Similarly, the INTERPSONDE VAP provides sounding profiles transformed onto a uniform time-height grid with 1-minute time resolution and 332 height levels up to 40 km, which counteracts the relatively lower time resolution schedule of four launches per day. The relative humidity and wind direction as a function of time and height are used here (Jensen et al., 2021).

### 2.2 KHGX WSR-88D Observations

The Next Generation Weather Radar (NEXRAD) network, a joint effort between the United States Departments of Commerce, Defense, and Transportation (Crum and Alberty, 1993), operates about 160 S-band Doppler dual-polarization radars across the country to provide consistent, high-resolution surveillance of weather phenomena for meteorologists and decision makers.

The Weather Surveillance Radar-1988 Doppler (WSR-88D) system responsible for monitoring the Houston, Texas and Galveston, Texas metropolitan areas is in Dickinson, Texas at 29º 28' 18.84" N, -95º 04' 43.82" E, and it is located 35 m above



mean sea level (Federal Meteorological Handbook No. 11, Part A, 2011). It is given the identifier KHGX. During the ESCAPE IOP, the KHGX WSR-88D observations were processed in real time to support the operation of the Multisensor Agile Adaptive Sampling (MAAS; Kollias et al., 2020; Lamer et al., 2023) algorithm implanted in two mechanically scanning radars to provide

convective cell tracking (Lamer et al., 2023; Kollias et al., 2024). The real-time, quality controlled KHGX WSR-88D observations from these MAAS activities are used here. The three-dimensional gridded KHGX radar data are also used to construct a 1.5 km above ground level (AGL) constant-altitude plan position indicator (CAPPI; Douglas, 1990), which is also used here.

**2.3 GOES-R ABI Observations**

The Geostationary Operational Environmental Satellite - R Series (GOES-R) is the National Oceanic and Atmospheric Administration's (NOAA) latest generation of satellites, providing users with continuous, high-resolution data to assist with weather forecasting, storm tracking, and atmospheric research (Geostationary Operational Environmental Satellite-R Series,

2019). The Advanced Baseline Imager (ABI) is one of the key instruments on GOES-R and provides radiance data in 16 spectral bands/channels ranging from the visible part of the electromagnetic spectrum to the near-infrared and the infrared. Of interest here, Channel 2 sits in the wavelength range of 0.59-0.69 μm and has an instantaneous field of view (IFOV) of 0.5 km (Schmit et al., 2005; Schmit et al., 2018), which is useful for studying fog, anvils, cumulus clouds, and ice/snow cover amongst phenomena. Channel 13 sits in the wavelength range of 10.1-10.6 μm and has an IGFOV of 2 km (Schmit et al., 2005; Schmit

et al., 2018) and is useful for studying atmospheric moisture, performing cloud identification and classification, estimating cloud top temperature and particle size, and characterizing surface properties (Lindsey et al., 2012). The reflectance data from Channel 2 and the brightness temperature data from Channel 13 are used here.

The ABI Cloud Mask (hereafter ACM) product combines nine of the 16 bands to provide a binary classification of "cloudy"

or "clear" for each pixel using spectral, spatial, and temporal signatures and has a horizontal resolution of 2 km (Heidinger and Straka III, 2013). There is also a four-level version of the mask which provides a classification of "cloudy", "probably cloudy", "probably clear", or "clear" for each pixel. This four-level cloud mask is used here.

**3 Methodology**


**3.1 Observational domain**

The analysis domain size is 250 km x 250 km, and it is centered at the location of the KHGX WSR-88D. Figure 1a shows the domain, with the KHGX site located southeast of the city of Houston and the AMF1 located northeast of the radar location



and southeast of the city. The KHGX and GOES-R data are gridded onto a horizontal Cartesian grid using a KD-Tree
       interpolation algorithm at a 500-m grid spacing (Bentley, 1975).

       An important methodology step is to consider limitations of the operational GOES-R and KHGX observations that will affect
       the cloud and precipitation analysis. A preliminary perusal of the GOES-R ABI Cloud Mask (ACM) indicated a noticeable
suppression of cloud detections in the grid points around the coastline of Houston compared to the surrounding areas. The
       surface properties (reflectance and emissivity) play a key role in the ACM performance, and this affects the ACM statistics in
       the areas where there is surface type transition (i.e. water to land) near the coastlines and water bodies over land. As a result,
       the grid points that correspond to these areas (white areas in Figure 1a) are ignored in further analysis of the GOES-R
       observations. Overall, 135,994 (87.5%) of the land grid points and 81,202 (85.9%) of the water grid points are used here. The
coverage, resolution and sensitivity of the KHGX radar observations depend on distance from the radar (Kollias et al., 2022).
       As a result, the 1.5 km CAPPIs show artifacts generated by the fact that only one elevation angle (the lowest) provided coverage
       at this height AGL. Thus, we have limited the KHGX radar analysis to a region 112.5 km from the radar location (Fig. 1b).
       Overall, 102,490 (65.9%) of the land grid points and 56,311 (59.6%) of the water grid points are used here.

**3.2 Comparison of KHGX WSR-88D, GOES-R ABI, and DOE ARM observations**

       The study of the cloud and precipitation spatiotemporal patterns presented in this study is based on the KHGX and GOES-R
       sensor fusion. The AMF1 provides a unique opportunity to evaluate the ability of the GOES-R ACM to detect clouds and the
       ability of the KHGX to detect light precipitation. Previous studies have assessed the cloud mask's skill using other satellite
measurements (McHardy et al., 2022), but, to our knowledge, a verification using ground-based measurements has never been
       conducted. An example of collocated KHGX, GOES-R and ARM observations is shown in figure 2. Figure 2 shows a time-
       height plot of KHGX observations over the AMF1 location using KHGX radar reflectivity data collected at different elevation
       angles. The ARSCL hydrometeor layer boundaries and the KAZR radar reflectivity are shown in figure 2b. Finally, the
       corresponding GOES-R ABI reflectance data from Channel 2 and the brightness temperature data from Channel 13 are shown
in figure 2c.

       Early on 23 July 2022, the GOES-R ACM accurately reports the presence of thin cirrus clouds identified in the ARSCL data
       (Fig. 2b). The cirrus clouds' low radar reflectivity and high altitude make them impossible to be detected with the KHGX
       WSR-88D due to the radar's limited sensitivity and the limit in the maximum height of the radar observations (dashed line in
Fig. 2a) imposed by the maximum elevation angle of the radar and its distance from the AMF1 site. Overall, there is good
       correspondence, in terms of cloud presence and duration, between the ACM and the cloud/precipitation and clear conditions
       as captured by the KHGX WSR-88D and AMF1 observations. The 'cloudy' periods show increases in radar reflectivity in
       both radars, increases in reflectance, and decreases in brightness temperature (Fig. 2). The comparison between the research-





grade radar in the ARSCL product and the KHGX WSR-88D also shows the value a WSR-88D can provide in representing a

cumulus cloud's morphology (width, cloud top height, etc.) despite its usage as surveillance. A more comprehensive

assessment of the ACM is provided in Appendix A.

The AMF1 measurements are also used to determine the lowest KHGX radar reflectivity value that can been used across the

analysis domain to detect the presence of precipitation in cumulus clouds. It is important to consider that WSR-88D radars are

sensitive not only to hydrometeors but also Bragg scattering and insects (Melnikov et al., 2011) and that the WSR-88D radar

sensitivity is inversely proportional to the square of the radar range.  The AMF1 surface disdrometer observations over the

four-month IOP are used to characterize the precipitation types and their frequency of occurrence at the ARM observatory.

Measured surface precipitation is observed 2.2% of the time (not shown). The maximum occurrence (5%) of measured

precipitation by the disdrometer occurs between 12 and 18 LT, with secondary maxima in late evening and before sunrise

hours (not shown).

The KHGX 1.5 km CAPPI radar reflectivity values over the AMF1 site are used to estimate the cumulative distribution of

KHGX radar reflectivity values between -10 dBZ and 60 dBZ (Fig. 3a).  Using the cumulative distribution of KHGX values

over the AMF1 site, a radar reflectivity threshold of 17 dBZ, where the frequency of occurrence of KHGX radar reflectivity

values and the frequency of occurrence of precipitation at the AMF1 (2.2%) as depicted by the disdrometer match, is identified

(Fig. 3a). The 17 dBZ radar reflectivity threshold is subsequently used to derive the diurnal variability of precipitation over

the AMF1 site, and it is compared to the diurnal variability of precipitation by the AMF1 disdrometer (Fig. 3b). Both have

similar timing for the local maxima overnight and early in the morning and the absolute maximum in the late afternoon, and

the magnitudes are similar, although the WSR-88D has more frequent precipitation at 21 Local Time (LT) and the disdrometer

has more frequent precipitation at 13 LT and 15 LT.

The 17 dBZ KHGX radar reflectivity threshold is suitable for matching the performance of a surface disdrometer to

characterize precipitation across the entire analysis domain. Furthermore, the 17 dBZ is well-above the minimum detectable

signal of the KHGX radar across the analysis domain. However, the disdrometer only accounts for precipitation that reaches

the surface and is above its detection limit (~0.1 mmhr$^{-1}$; Angulo-Martinez, 2018). To account for lighter precipitation that

does not reach the surface (i.e., virga) or is below the rainfall rate detection limit of the disdrometer, the combined ARSCL

profiling radar and lidar observations are used. Using the four-month long ARSCL record, the frequency of detecting

hydrometeors as low as 160 m AGL was estimated (4.5%).  Estimates of the lifting condensation level at the AMF1 site (not

shown) indicated that no cloud bases are expected so close to ground; thus, the assumption is made that the observed

hydrometeors are falling drizzle and raindrops. The histogram of the KAZR radar reflectivity values during these occurrences

is shown in Fig. 3c.  The distribution peaks between 0 and 20 dBZ. The ARSCL hydrometeor mask detection near the surface

is used as a more sensitive criterion for the detection of precipitation at the AMF1 site. The corresponding KHGX radar



reflectivity threshold needed to match the 4.5% occurrence of precipitation is 4 dBZ. The shapes of the diurnal cycle profiles of precipitation occurrence in the WSR-88D radar and ARSCL data show good agreement, while the magnitudes of the local

and absolute maxima show slight differences, i.e. the absolute maxima at 13 LT are about 2% less in the WSR-88D data than the ARSCL data (Fig. 3d).

The KHGX radar's sensitivity reduces with the square of the radar's range. If we want to use the 4 dBZ radar reflectivity threshold, we must first confirm that it is above the KHGX radar's minimum detectable signal (as depicted in the 1.5 km AGL

CAPPI data) and that it is above any Bragg scattering returns. At the furthest range considered in this study, the minimum detectable signal is close to 5dBZ. Given that this is slightly above our current precipitation threshold, we adjust our threshold to 10 dBZ instead of 5 dBZ to avoid choosing one so close to the minimum radar reflectivity the KHGX WSR-88D can observe at far ranges. It is important to note that a 10 dBZ precipitation threshold for the KHGX radar corresponds to a rainfall rate below 0.088 mmhr$^{-1}$. The conversion of the KHGX radar reflectivity to rainfall rate (R) was done using the climatological

relationship used for this regime by the National Weather Service (R = $300Z^{1.4}$).

Finally, when using the radar reflectivity threshold of 10 dBZ, we find that the data still shows artifacts from nonmeteorological scattering, especially in the late evening and overnight hours. As in Lamer et al. (2023), we use a vertically integrated liquid (VIL) threshold to suppress these signals and keep only meteorological scatterers. We estimate VIL using the Marshall-Palmer

drop size distribution (Marshall and Palmer, 1948) assumptions:

$$VIL = \sum_{i=0}^{top} 3.44 \times 10^{-6} [\frac{Z_i + Z_{i+1}}{2}]^{4/7} \Delta h$$

where Z stands for radar reflectivity (in dBZ) and $\Delta h$ stands for the depth of the layer between consecutive grid levels (in m). Here, we use the same threshold of 0.1 kg m$^{-2}$ as Lamer et al. (2023). Every pixel must then have a radar reflectivity greater than 10 dBZ and a VIL greater than 0.1 kg m$^{-2}$ to be kept in the dataset.


### 3.3 Shallow and deep cloud/precipitation classification

An example of the cloud and precipitation classification during an ESCAPE IOP on 2 June 2022 at 11:22:36 LT is shown in Figure 4. The GOES-R ACM data are used to define clear and cloudy conditions (Fig. 4b). The GOES-R Channel 13 brightness

temperature data ($T_{B13}$) in Fig. 4a show widespread shallow cloudiness across the domain, except a small region south and east of Galveston Bay where colder $T_{B13}$ are observed. The $T_{B13}$ are used to further classify the clouds identified in the GOES-R ACM data as warm phase only if $T_{B13} > -5$ °C, in which it is reasonable to assume there is no ice present, and as mixed-phase only if $T_{B13} < -5$ °C. The cloud top height and temperature are strongly related; thus, the warm phase only clouds are hereafter called shallow convection and the mixed-phase clouds are called deep convection (Figs. 4c-d). In addition, the gridded KHGX



CAPPI data are used to classify the detected clouds as precipitating or not (Fig. 4f-h). We use the radar reflectivity and VIL thresholds described above, and we classify each pixel as shallow or deep by comparing the radar echo top height at that pixel to the median height of the -5 °C level (5.79 km; not shown) derived from the AMF1 soundings, where a radar echo top height less than 5.79 km is shallow and greater than 5.79 km is deep. At this instance in time, the shallow clouds over the Gulf (Fig. 4c) are not precipitating (Fig. 4g) from the radar's perspective, while more of the shallow clouds along the coastline and over

land are in fact precipitating. There is a prominent cold top feature east of Galveston Bay (Fig. 4d) with some deep precipitation associated with it (Fig. 4h). Otherwise, this instance in time is defined by more shallow cloud and precipitation features, with abundant activity along the coastline.

## 4 Results


### 4.1 Domain-averaged diurnal cycle of clouds and precipitation

The gridded KHGX radar and GOES-R data products are used to estimate the fraction of the analysis domain covered by warm (shallow) and cold (deep) cloud tops. We calculate the number of shallow cloud, deep cloud, shallow precipitation, and deep

precipitation pixels and divide by the total number of pixels in the domain used for each instrument. The domain-average fractions are estimated approximately every five minutes, and their median value within an hour is used to derive the diurnal variability of shallow and deep clouds throughout the four-month TRACER and ESCAPE IOP period (Fig. 5). The maximum hourly median shallow cloud fraction in the dataset is 84.2% at 9 LT on 8 June; meanwhile, the maximum hourly median deep cloud fraction in the dataset is 100% at 15 LT on 4 September. Of the 121 days that are not missing data, 37 (30.6%) have

their maximum hourly median shallow cloud fraction occur between 0-5 LT, 13 (10.7%) have their maximum occur between 6-11 LT, 31 (25.6%) have their maximum occur between 12-17 LT, and 40 (33.1%) have their maximum occur between 18-23 LT. Similarly, of the 112 days that are not missing data and that have hourly median deep cloud fractions above 0% all day, 19 (17%) have their maximum hourly median shallow cloud fraction occur between 0-5 LT, 12 (10.7%) have their maximum occur between 6-11 LT, 44 (39.3%) have their maximum occur between 12-17 LT, and 37 (33%) have their maximum occur

between 18-23 LT. Indicated by these statistics, the morning hours of 6-11 LT experience the maximum hourly median shallow and deep cloud fractions least frequently, and the maximum hourly median shallow cloud fraction occurs most frequently in the evening while the maximum hourly median deep cloud fraction occurs most frequently in the afternoon. Overall, the domain-averaged fractional coverage of shallow clouds indicates that they are more frequently observed in the domain throughout the IOP period compared to deep clouds. The domain-averaged shallow cloud fraction is always above 0% while

no deep clouds are observed in the domain on nine separate days. Likewise, the maximum hourly median shallow cloud fraction is below 5% only in four days, including one day in July and three days in September (Fig. 5a), whereas there are 28 days when the maximum hourly median deep cloud fraction is below 5%, including six in June, seven in July, and 16 in September (Fig. 5b). August has the most widespread shallow and deep cloud activity, with 22 days and 17 days, respectively,



having maximum hourly median cloud fractions exceeding 50%. July has the fewest days where the maximum hourly median
deep cloud fraction exceeds 50% (4 days) while September has the fewest days where the maximum daily median shallow
cloud fraction exceeds 50% (5 days).

The diurnal variability of the domain-averaged shallow and deep precipitation is estimated, too (Fig. 6). The maximum hourly
median shallow precipitation fraction in the dataset is 9.3% at 15 LT on 15 July; meanwhile, the maximum hourly median
deep precipitation fraction in the dataset is 63.7% at 17 LT on 3 September. Of the 110 days that are not missing data and that
have hourly median shallow precipitation fractions above 0% all day, 19 (17.3%) have their maximum hourly median shallow
precipitation fraction occur between 0-5 LT, 36 (32.7%) have their maximum occur between 6-11 LT, 42 (38.2%) have their
maximum occur between 12-17 LT, and 13 (11.8%) have their maximum occur between 18-23 LT. Similarly, of the 95 days
that are not missing data and that have hourly median deep precipitation fractions above 0% all day, 9 (9.5%) have their
maximum hourly median shallow precipitation fraction occur between 0-5 LT, 16 (16.8%) have their maximum occur between
6-11 LT, 54 (56.8%) have their maximum occur between 12-17 LT, and 16 (16.8%) have their maximum occur between 18-
23 LT. Of the four variables thus far, the maximum hourly median deep precipitation fraction shows the strongest preference
for a period of the day, in this case 12-17 LT, and the maximum hourly median shallow precipitation fraction occurs most
frequently then, too. Overall, the precipitation activity is less widespread than the cloud activity. For 68 days (more than half
of the total days), the maximum hourly median shallow precipitation fraction never exceeds 5%, which includes 27 days in
June, 26 days in July, 11 days in August, and four days in September. Similarly, for 81 days (two-thirds of the total days), the
maximum hourly median deep precipitation fraction never exceeds 5%, which includes 24 days in June, 23 days in July, 12
days in August, and 22 days in September. Finally, the deep precipitation in general is far more widespread than the shallow
precipitation, as only 7 days have maximum hourly median shallow precipitation fractions greater than 5% while 23 days and
10 days have maximum hourly median deep precipitation fractions greater than 10% and 25%, respectively.

## 4.2 Shallow cloud fraction variability

A k-means clustering approach (Lloyd, 1982) is applied to the high-resolution data to identify the dominant diurnal modes of
shallow cloudiness across the analysis domain. A principal component analysis (Pearson, 1901) of the same data indicated that
the first four principal components (or modes) account for over 90% of the variability in the data. As a result, k is set to four
in the k-means algorithm (Throndike, 1953), and we use the 121 median diurnal patterns of shallow cloud fraction as the input.
Each day (September 13th is not classified because of missing data) is assigned to one of four clusters. Finally, for each cluster,
we gather all the data and calculate the median shallow cloud fraction and the shallow cloud fraction interquartile range every
hour.





The diurnal variability of the four dominant modes of median shallow cloudiness and their interquartile ranges every hour for each of the four clusters is shown in figure 7. The four modes have distinct characteristics in terms of magnitude, shape, and timing of maxima and minima. Cluster 1 (C1) is the largest with 48 days grouped in this mode. C1 represents days with the lowest amount of shallow cloudiness, with the median shallow cloud fraction never exceeding 10% (Fig. 7a). The median fraction is at its minimum of 1.4% at 23 LT; it begins growing at 4 LT and reaches its maximum of 9.2% at 15 LT in the afternoon. The 21 days belonging to Cluster 2 (C2) have higher shallow cloud fractions and exhibit the opposite diurnal cycle compared to C1 (Fig. 7b). In C2, the median shallow cloud fraction is at its highest value (57.9%) at 22 LT in the evening while reaching its absolute minimum of 34.1% at 16 LT in the afternoon. The 22 days in Cluster 3 (C3) have their diurnal absolute maximum of 53.5% occurring at 2 LT, and the median shallow cloud fraction decreases steadily for the rest of the day from 51% at 5 LT to 14.8% at 23 LT (Fig. 7c). Finally, although the magnitude of the median shallow cloud fraction is larger, the 30 days in Cluster 4 (C4) behave similarly to C1 in that there is steady growth starting in the late morning and ending in the afternoon (Fig. 7d). The median shallow cloud fraction reaches 26.1% at 13 LT and then persists between 27% and 30% for the rest of the day. Monthly calendars corresponding to the four months of the campaigns and the cluster each day belongs are shown in Appendix B.

### 4.3 Cloud and precipitation fraction variability

The diurnal variability of shallow cloudiness and associated precipitation is affected by deep convective clouds, their mesoscale organization, and their impact on boundary layer organization. Using C1-4, the diurnal analysis is expanded to include the shallow precipitation fraction and the deep cloud and precipitation fractions. Their median hourly values, as well as the interquartile ranges, for C1-4 are shown in figure 8. C2 and C3 have significantly more fractional coverage of clouds with cold tops, with their diurnal maxima of 46.6% at 16 LT and 41.3% at 16 LT, respectively, and the median deep cloud fraction is above 0% all day (Figs. 8b-c). Furthermore, deep cloud fraction is higher than shallow cloud fraction between 14-19 LT in C2 and between 15-18 LT in C3. In all clusters (Figs. 8a-d), the hourly median deep cloud fraction consistently reaches its diurnal maximum in the late afternoon (0.1% at 17 LT in C1, 46.6% at 16 LT in C2, 41.2% at 16 LT in C3, and 13.2% at 16 LT in C4). C1 and C4 show a similar pattern to one another: the shallow cloud fraction begins to increase in the late morning and then the deep cloud fraction begins to increase 2-3 hours later (Figs. 8a and 8d). Finally, the maximum hourly median shallow cloud fraction occurs six hours after the maximum hourly median deep cloud fraction in C2, while the maximum hourly median shallow cloud fraction occurs 14 hours earlier than the maximum hourly median deep cloud fraction in C2 (Figs. 8b-c).

The domain is covered by far less precipitation; the median shallow and deep precipitation fractions never exceed 5% regardless of the cluster (Figs. 8e-h). Comparing the cloud and precipitation data, the median deep precipitation fractions in C1, C2, and C4, like the median deep cloud fractions, attain their diurnal maxima in the late afternoon (0.01% at 17 LT in C1,





2.3% at 16 LT in C2, and 0.85% at 15 LT in C4). Meanwhile, in C3, the median deep precipitation fraction reaches its maximum earlier in the day than the median deep cloud fraction does (8 LT compared to 16 LT). The median shallow cloud and precipitation clusters in all four clusters do not behave similarly and therefore have different maxima timing. C2 and C3 again have periods when the median deep precipitation fraction exceeds the median shallow precipitation fraction, between 8-

19 LT and 6-17 LT, respectively (Figs. 8f-g). C4 even has this feature between 15-18 LT (Fig. 8h).

**4.4 Land versus Ocean**

About 40% and 60% of the analysis domain is covered by ocean and land, respectively. Due to the prominent sea and land
breeze circulations in Houston (Kocen, 2013) and the differences in convection over land and ocean, we repeat the analysis for the land and ocean portions of the analysis domain (Figs. 9 and 10). There are some noticeable differences in the timing and magnitude of the median shallow cloud fraction over land and over water. C1 shows a larger maximum median shallow cloud fraction occurring over water (10% at 8 LT) five hours earlier than the one occurring over land (7.5% at 13 LT; Figs. 9a and 9e), and, while both profiles see their largest values in the latter half of the day, the median shallow cloud fraction over
land in C4 reaches its maximum of 28.7% at 15 LT while the median shallow cloud fraction over water in C4 reaches a larger maximum of 44.2% later in the day at 22 LT (Figs. 9d and 9h). Although the timing of the maxima in the shallow cloud fractions over land and over ocean is relatively similar in C2 and C3 (22 LT to 23 LT and 2 LT to 5 LT, respectively; Figs. 9b-c and 9f-g), the magnitudes over water are greater, meaning, across all four clusters, the median shallow cloud fraction over water is consistently greater than it is over land. On the other hand, we see that the maxima of the median deep cloud fractions
over land and over water are similar in timing (a one-hour difference in C2, a two-hour difference in C3, and no difference in C4), but the magnitudes over land are higher than they are over water across all four clusters (0.11% to 0% in C1, 55.4% to 41.2% in C2, 38.6% to 33.1% in C3, and 16.6% to 5.1% in C4; Fig. 9). Breaking down the two surface types also allows us to see that the periods where the deep cloud fractions exceed the shallow cloud fractions in C2 and C3 from figure 8 persist regardless of land or ocean (Figs. 9b-c and 9f-g).


Precipitation data behaves differently according to surface types. The median deep precipitation fraction over water reaches its diurnal maximum before the median deep precipitation fraction over land does in C2 and C4 (7 LT to 16 LT and 8 LT to 15 LT, respectively; Fig. 10), and the same goes for the shallow precipitation fractions in C1, C2, and C4 (6 LT to 17 LT, 7 LT to 13 LT, and 6 LT to 15 LT, respectively). On the other hand, C3 shows that the median deep and shallow precipitation
fractions over land peak before they do over water (8 LT to 16 LT and 8 LT to 20 LT, respectively; Figs. 10c and 10g). However, regardless of cluster, the maximum median deep and shallow precipitation fractions over land are higher than those over water. Also, the shallow and deep precipitation over land has a shift in timing depending on the clusters: the median shallow precipitation fraction peaks two hours before the median deep precipitation fraction in C2 (Fig. 10b), at the same time in C3 (Fig. 10c), and at the same time in C4 (Fig. 10d). Comparing the cloud and precipitation data over land and over water,



there is a larger discrepancy in the timing of the maximum deep precipitation and cloud fractions over land in C3 (8 LT to 16 LT; Fig. 9c and Fig. 10c) than there is in C2 and C4 (18 LT to 16 LT and 18 LT to 15 LT, respectively). C1's median deep precipitation fraction over land never exceeds 0%.

Lastly, we calculate the diurnal cycle of the percentage of clouds and precipitation that are classified as shallow over land and
over water in Clusters 1 and 4 in figure 11. Given these two feature the lowest spatial coverage of deep clouds and precipitation, we can begin to evaluate what controls shallow clouds' ability to transition into deep clouds. Over land in C1, between 41.7-66% of clouds are shallow, with the minimum occurring at 8 LT and the maximum occurring at 13 LT, and between 22.8-55.5% are precipitation is shallow, with the minimum occurring at 19 LT and the maximum occurring at 6 LT (Fig. 11a). Over water in C1, the percentage of clouds that are shallow is similarly between 42.1-67%, only the minimum's and maximum's
timings are reversed at 18 LT and 9 LT, respectively (Fig. 11c). The percentage of precipitation that is shallow shows far more variability over water and it does over land, as the minimum is 9.3% at 15 LT and the maximum is 75.6% at 0 LT. Over land in C4, the percentage of clouds that are shallow is consistently higher than it is in C1, as 20 different hours feature a percentage greater than 60% in C4 compared to five different hours in C1 (Figs. 11a-b). The cloud and precipitation percentages in C4 over land also a similar diurnal pattern: a maximum attained in the morning hours between 8-11 LT followed by a steady
decrease until the absolute minimum is reached after 4 LT (Figs. 11b). The precipitation percentage is also more variable over water than it is over land in C4, having a minimum of 7.5% at 19 LT and a maximum of 83.8% at 23 LT (Fig. 11d). Finally, we will note a striking difference between land and water in C4. The percentage of clouds that are shallow over water is steadily between 70.8-85.3%, and as noted already, the percentage of clouds that are shallow over land features a large decrease between morning and late afternoon. More exactly, the percentage for clouds over land starts at 84.4% at 11 LT and reaches
50% by 18 LT; meanwhile, in that same time frame, the percentage over water starts at 77.4% and ends at 76.7%. Some forcing mechanism over land is causing more shallow clouds to transition into deep clouds that does not exist over water. It is likely we do not see this pattern in C1 because deep cloud and precipitation activity is minimal already.

## 4.5 Meteorological variability in the clusters


The INTERPSONDE VAP, which provides 1-minute resolution profiling data of temperature, relative humidity, and wind speed and direction, is used to describe the meteorological conditions in these clusters. The median time-height cross sections of relative humidity and wind direction variables for the four clusters are shown in figure 12. All clusters show a similar diurnal cycle below 1 km: relative humidity values greater than 80% between 0-6 LT and then a steady drop until the minimum is
reached in the early afternoon (Figs. 12a-d). For example, C1 has the lowest relative humidity during these afternoon hours, with a value of 54.9% at 8 m at 13 LT (Fig. 12a). At the same height, C2 reaches 67% between 11-13 LT, C3 reaches 63.5% at 11 LT, and C4 reaches 60.1% at 13 LT (Figs. 12b-d). All four clusters feature a humidity inversion between 12-18 LT, and the region of higher relative humidity (compared to the surface) exists between 1-2 km. The most noticeable difference across



the clusters is the depth of the moisture layer. C1 is noticeably drier (RH < 20%) above 6 km compared to the other three

clusters (Fig. 12a), and this, combined with the dryest surface conditions in the afternoon, may help explain the lack of deep cloud and precipitation presence on these days. This region above 6 km is also dryer in C4 compared to C2-3. Overall, C2 and C3 feature deeper moisture profiles, and there is a clear correlation between this and the larger cloud and precipitation fractions seen on these days.

The four clusters show a similarity near the surface in the wind direction data, too. The first nine to ten hours of the day feature a wind direction coming from the south or southwest. This persists after 11-12 LT between 1-2 km while the wind direction shifts and is coming from the southeast below 1 km (Figs. 12e-h). This veering wind profile is indicative of warm air advection, and, given the location of the AMF1 (Fig. 1), a southeast wind would mixing marine air from Houston – Galveston Bay (HGB) with continental air from the southwest wind above it. Like the relative humidity profiles, the upper atmosphere is where the

glaring differences in the clusters appear. There are north-westerly winds above 10 km in C2-3 (Figs. 12f-g) while most of the wind directions above 4 km in C4 are coming from the east or northeast (Fig. 12h).

## 4.6 Spatial variability in clusters 1 and 4

C1 and C4 are the most frequently occurring modes and represent atmospheric conditions with minimal to no deep clouds and precipitation. This provides an opportunity to examine the shallow clouds and precipitation present in these modes, and so we will focus on these two clusters for the remainder of the study. The domain-averaged diurnal variability has indicated that influence of the land-ocean contrast is important in determining the diurnal cycle. Here, the spatial variability in these two clusters during daytime (09:00 – 19:00 LT) is investigated. At every pixel, we calculate the number of shallow cloud

occurrences and divide by the number of total data points. The spatiotemporal variability of shallow cloud frequency every 2 hours from 9:00 to 19:00 LT for C1 and C4 is shown in figure 13. In the morning at 9 LT, shallow clouds in C1 are more frequently occurring over water, and a sharp gradient in shallow cloud frequency is observed across the coastline (Fig. 13a). The maximum in shallow cloud frequency (15-20%) over the ocean during early morning hours is consistent with the diurnal variability of the oceanic shallow cloud fraction (Fig. 10e) that maximizes in early morning hours. Later in the day, as a

convective boundary layer forms over land, shallow clouds frequency begins to increase over land as well. The maximum of the C1 shallow cloud fraction is observed initially near the coast and gradually shifts further inland, aided onshore flow while the penetration of the stable marine boundary layer air suppresses shallow cloud frequency near the coastline. In the meantime, shallow cloud frequency over the ocean gradually decreases and reaches a minimum after 17 LT (< 5%; Figs. 13e-f).

C4 is associated with much higher shallow cloud frequency (Figs. 13g-l). In the morning at 9 LT, higher shallow cloud frequency is observed over water than over land as in C1 (Fig. 13g). In the transition from late morning to early afternoon (Figs. 13h-i), the contrast in shallow cloud frequency between the ocean and land disappears, and shallow cumulus fields are





observed across the domain with 25-35% frequency. In C4, some noticeable spatial patterns are observed at 15 and 17 LT (Figs. 13j-k), and these patterns are not associated with the coastline and the surface type transition. Higher shallow cloud
frequencies (25-40%) are observed east of HGB while, west of the HGB area, lower shallow cloud frequencies (10-25%) are observed (Figs. 13j-k). At 15 LT, the maximum shallow cloudiness is observed over land west of HGB, and, in the later afternoon and evening hours, the maximum shallow cloudiness frequency is observed over water unlike in C1 (>40%; Figs. 13k-l).

Overall, the spatiotemporal variability of the shallow clouds fractional coverage is significant. C1 represents more suppressed atmospheric conditions, and surface properties (land versus ocean) and boundary layer stability likely explain most of the observed variability. C4 represents conditions that can support higher shallow cloud frequency. More complex processes and surface/atmospheric properties are responsible for the observed spatiotemporal patterns. The area east of the HGB is home to shipping lanes and most of the refineries and industrial complexes while the area west of HGB has less polluted air (Kollias et
al., 2024). No comprehensive aerosol data are available that can match the spatiotemporal sampling of the operational sensors used in this study; thus, no attempt is made to correlate the shallow convective clouds properties with aerosol conditions.

The corresponding spatiotemporal variability of shallow cloud top heights every 2 hours from 9:00 to 19:00 LT for C1 and C4 is shown in figure 14. Using the soundings launched from the AMF1 site, we convert the $T_{B13}$ of the cloudy pixels to cloud top
height, and we calculate the median cloud top height in 10 km x 10 km boxes every hour for each cluster. During early morning (9 LT), C1 shallow clouds have higher cloud tops (>2 km) over land (Fig. 14a). During 11 and 13 LT, the cloud top heights over land are lower (Figs. 14b-c). One noticeable feature is that higher cloud top heights are observed in the area east of HGB compared to the area west of HGB. This trend persists and amplifies as we progress through the day (Figs. 14d-e). On the other hand, C1 shallow cumulus over ocean have low cloud tops (<1 km) and maintain these low cloud tops throughout the day.
The higher cloud top heights east of HGB are also present in C4 (Figs. 14h-k). While shallow clouds get deeper west of the HGB in early and later afternoon (15-17 LT), the corresponding cloud top heights east of the HGB are higher (Figs. 14j-k). By 19 LT, the contrast between the sides of HGB muddies, and this is consistent in both clusters (Figs. 14f and 14l). The presence of shallow clouds with higher cloud top heights east of the HGB area is persistent and coherent in time and space while also detectable in both C1 and C4. It suggests that, in this area, shallow clouds have stronger updrafts possibly driven by surface
and aerosol properties.

Finally, the combined KHGX and GOES-R cloud and precipitation data are used to estimate the spatiotemporal variability of the shallow cloud-to-precipitation ratio for C1 and C4 in figure 15. For each 10 km x 10 km box, we divide the total number of shallow cloud occurrences (precipitating and non-precipitating) by the total number of shallow precipitation occurrences
every hour for each cluster. At 9 LT in C1, shallow clouds over ocean are more likely to precipitate than their counterparts over land, but, overall, the cloud-to-rain ratios are very low (Fig. 15a). A prominent band of very low cloud-to-precipitation





ratios is observed at 15 LT on both sides of HGB (Fig. 15d), and the band of low cloud-to-precipitation ratios coincides with the area of high shallow cloud frequency in figure 11d. At 17 LT, a band with very high cloud-to-precipitation ratios is observed along the coastline while a less pronounced band of low cloud-to-precipitation ratios in observed further inland (Figs. 15e). At

19 LT, little precipitation is observed in the domain (Fig. 15f). This signal is not observed in C4, as none of the six time periods examined have a pronounced area of low cloud-to-precipitation ratio. Higher occurrences of clouds with higher cloud tops are the key feature of C4.

## 5 Summary and Discussion


In this study, we characterize the spatiotemporal variability of clouds and precipitation properties in the Houston area during the ESCAPE and TRACER field experiments. The Houston area is a natural laboratory for the study of convective clouds. However, there are several factors that could influence the spatiotemporal variability of shallow clouds specifically such as sea breeze circulations, varying aerosol conditions, surface heterogeneities, and cold pools from deep convection. These factors

can operate in isolation or at the same time to influence the mesoscale organization of shallow cumulus clouds, their depth, and propensity to precipitate. Any observational effort that aims to disentangle these effects should be based on a large data sample. Here, operational satellite and radar datasets are used to provide statistics on cloud fraction, cloud top heights, and precipitation occurrence over a large domain. A novel aspect of the proposed methodology is that we utilize the state-of-the-art instrumentation at the TRACER AMF1 site to validate and constrain the operational datasets. We evaluate the performance

of the GOES-R cloud mask product and estimate a radar reflectivity threshold for the KHGX that is applicable in the entire domain and can detect the presence of light rainfall rate ($< 0.1$ mmhr$^{-1}$).

Throughout the four-month period, we identify all shallow and deep clouds and precipitation, and we establish the four main diurnal patterns of shallow cloud fraction in the domain. We quantify the timing and magnitude of the cloud and precipitation

occurrences while also noting the differences between land and water. Mainly, we find the cloud coverage is larger than the precipitation coverage, and we show that the deep clouds' coverage maximizes in the afternoon in all four modes while three modes have deep cloud and precipitation coverages maximizing at the same time. Shallow cloud coverage is higher over water than it is over land while the shallow precipitation coverage shows the opposite, and deep cloud and precipitation coverage is higher over land across all four modes.


One of the reasons for identifying the different modes of diurnal shallow cloud fraction variability is to identify time periods that have limited deep cloud activity. Clusters 1 and 4 fit this description and, at the same time, are the most populous clusters; thus, we have a healthy sample size to study their spatiotemporal variability. The rationale of the proposed approach is that, by focusing on C1 and C4, we could potentially isolate the role of the sea breeze circulation on shallow cloud properties and

at the same time investigate any other spatial patterns that can affect shallow cloud properties. For example, C1 contains days



with negligible amounts of precipitation and low shallow cloud fraction, and C4 contains days with relatively higher shallow cloud fraction. When considering the spatiotemporal variability of the cloud top heights, the presence of shallow clouds with higher cloud top heights east of the HGB is persistent and coherent in time and space. It suggests that, in this area, shallow clouds have stronger updrafts which may be possibly driven by surface and aerosol properties. The spatiotemporal variability

of the cloud-to-precipitation ratios show a prominent band of very low cloud-to-precipitation ratios in C1 in the early afternoon. This band of low cloud-to-precipitation ratios coincides with the area of maximum shallow cloudiness, too. Later in the day, very high cloud-to-precipitation ratios are observed along the coastline while less pronounced low cloud-to-rain ratios are observed further inland

Because of our large sample sizes, we can identify robust signals in the domain like these that warrant future investigation. We hypothesize that the low cloud-to-precipitation ratios in the afternoon in C1 is related to the sea breeze, but without a comprehensive thermodynamic analysis of variables like virtual potential temperature, we cannot say that with certainty. Combine that analysis with a radar and satellite-derived boundary identification, and we could empirically identify the presence of sea breeze conditions for each day and clarify how much of an impact sea breezes have on the signals in C1. It appears sea

breeze conditions play more of a role in determining shallow cloud properties in C1, but the forcing mechanism over land that causes more shallow clouds to become deep clouds in C4 could be the sea breeze, too. We do not expect every day in C1 and C4 to have sea breeze conditions, but based on the composites we have, the days that do may be driving the signals.

Likewise, more comprehensive aerosol data is needed to clarify the possible warm phase invigoration signal we see in the

shallow cloud top heights. Data on aerosol number concentration and composition as well as supersaturation provided at varying heights and with high time-resolution would help with attribution. Similarly, we only provide meteorological measurements over one location, albeit in the center of the domain, and we do not know how representative that one profile is. More observations along with reanalysis data could better identify possible causation between meteorological properties, like relative humidity, and our cloud and precipitation statistics.


Finally, we believe this analysis is useful and informative for the modelling studies conducted on cases from the ESCAPE and TRACER campaigns. A simulation's ability to accurately recreate the cloud and precipitation fields in the domain for all cases is an important first step in the process that would then include deciphering the impact of aerosols, both natural and anthropogenic, and mesoscale circulations like sea breezes on the cloud and precipitation themselves. We also believe that this

analysis helps provide a first glimpse into days with widespread deep convection that could be scientifically interesting. Although we did not focus on deep clouds in this study, days falling in Clusters 2 and 3 especially could provide ample opportunities to study deep convective cloud properties and the shallow-to-deep transition.

**6 Appendices**






## 6.1 Appendix A

To provide a more comprehensive assessment of the cloud mask's skill, we use the following skill metrics:

$$\text{Accuracy} = \frac{\text{True Positives (\#)} + \text{True Negatives (\#)}}{\text{True Positives (\#)} + \text{False Positives (\#)} + \text{False Positives (\#)} + \text{True Negatives (\#)}}$$

$$\text{Precision} = \frac{\text{True Positives (\#)}}{\text{True Positives (\#)} + \text{False Positives (\#)}}$$

For our purposes, we use the ARSCL data as the truth and the cloud mask data at the grid point over AMF1 as the prediction. We then consider a true positive to be an instance when both datasets report cloudy conditions, a false positive to be an instance
when the cloud mask reports cloudy conditions and ARSCL reports clear conditions, a false negative to be an instance when the cloud mask reports clear conditions and ARSCL reports cloudy conditions, and a true negative to be an instance when both products report clear conditions.

One final consideration to make is the different temporal resolutions of the ARSCL and cloud mask datasets; the cloud mask
provides data every five minutes and the ARSCL product provides data every few seconds. Initially, in the five minutes leading up to the cloud mask data point, if ARSCL has at least one cloud base height measurement, we designate the truth as cloudy and compare it to the prediction. The accuracy and precision of the cloud mask across the four-month period are 75% and 98%, respectively, or the leftmost data points in Figure A1a. The exceptional precision means the cloud mask is not false reporting cloudy conditions as clear conditions, but the accuracy leaves much to be desired. To improve the accuracy, we use
stricter criteria: a greater fraction of the five minutes leading up to the cloud mask measurement must have cloud base height measurements in the ARSCL data. We incrementally increase this fraction until we reach 1 (or 100%), meaning there must be cloud base height measurements for the entire five-minute period in the ARSCL data for the truth to be designated as cloudy. Figure A1a displays the accuracy and precision as a function of this percentage. At a value of 100%, the precision drops to about 90%, which is still satisfactorily high. The accuracy shows marked improvement, reaching 86%. We use a percentage
of 100 for the rest of the analysis.

We calculate the accuracy and precision as a function of time in Figure A1b during the four months. The accuracy stays between 81.9% and 91% throughout the day with an absolute maximum at 8 LT and a local maximum at 15 LT, and the precision is highest between 0-3 LT and 20-23 LT while experiencing an absolute minimum of 71.7% at 12 LT (Fig. A1b).
Heidinger and Straka III (2013), using CALIPSO measurements to validate the cloud mask, found a similar diurnal discrepancy, as they showed the cloud mask had a probability of detection (accuracy) of 93.9% and a false cloud detection of



4.6% over land during the day and a probability of detection (accuracy) of 89.5% and false cloud percentage of 2.2% over land at night. Jimenez (2020) compared the cloud mask to CALIPSO measurements over the contiguous United States and also found lower performance during the daytime in summer due to misdetections of clear sky conditions. Both studies showed
lower performance with low cloud as well, so lastly, we examine the accuracy as a function of cloud base height. We calculate the median cloud base height of each five-minute period and test the cloud mask's accuracy as a function of height. Figure A1c indicates the accuracy is greater than 80% for median cloud bases less than 8 km; the accuracy decreases rapidly with median cloud base heights greater than this. These encouraging results suggests the cloud mask dataset is a feasible one for cloud research, and the improved performance for clouds closer to the surface in our analysis may be due to using ground-
based measurements for validation instead of satellite-based measurements used in previous studies.

**6.2 Appendix B**

Monthly calendars corresponding to the four months of the ESCAPE and TRACER campaigns and the cluster each day belongs
are shown in Figure B1. June has a representative mix of the four clusters, with 12 C1 days, five C2 days, six C3 days, and seven C4 days (Fig. B1), and July is similar, with 14 C1 days, two C2 days, five C3 days, and 10 C4 days. August and September shift the pattern, as August has the least C1 days (two of 31 total days) of the four months while C1 days dominate September (20 of 30 total days). To see the transition from consistent domain-wide shallow cloudiness in August to far less in September leads us to believe we are observing the seasonal transition from summer to fall, which also marks the end of the
convective season in Houston (Fridlind et al., 2019).

**7 Code availability**

All codes used in this paper are available upon request.

**8 Data Availability**

The datasets from the TRACER AMF1 site can be found on the ARM Data Center (https://adc.arm.gov/discovery/#/results/iopShortName::amf2021tracer). The operational radar and satellite datasets are
available through the National Centers for Environmental Information.

**9 Author Contributions**





ZM and PK designed the methodology, and ZM performed the analysis. ZM and PK prepared the paper, with contributions
and feedback from PB and MO. BPT curated the datasets used in the analysis. PK and MO acquired the funding for this work,
and PK served as the PhD advisor to ZM while MO served as a PhD committee member to ZM.

**10 Competing Interests**

The authors declare that they have no conflict of interest.

**11 Acknowledgements**

We would like to thank Crameri (2021) for their colour-vision deficiency friendly and perceptually uniform colour maps used
in various figures of this manuscript. We would also like to acknowledge the data support provided by the Atmospheric
Radiation Measurement (ARM) Program sponsored by the U.S. Department of Energy. Finally, we would like to thank the
National Science Foundation and the U.S. Department of Energy for supporting this work.

**12 Financial Support**


ZM, PK, BPT, and MO were supported by the National Science Foundation (Award AGS 2019932). MO was also supported
by the U.S. Department of Energy, Atmospheric System Research (Contract DE-SC0021160). PK was also supported by the
U.S. Department of Energy, Atmospheric System Research (Contract DE-SC0012704).

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

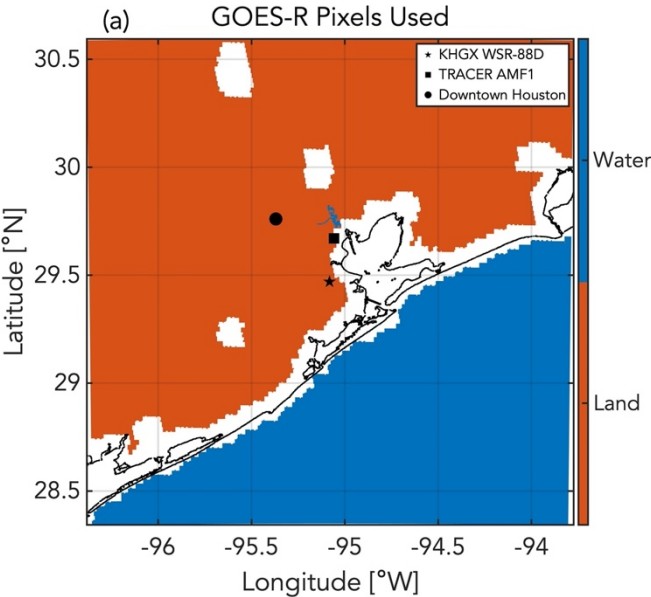
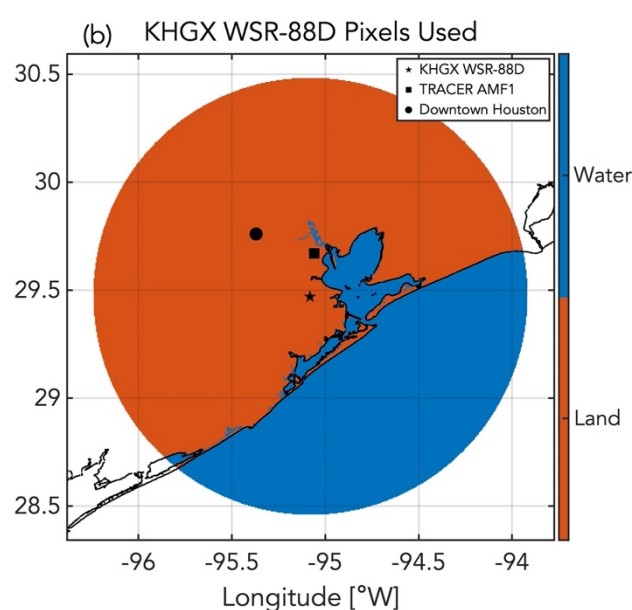



**Figure 1: For the 250 km x 250 km grid used in this study, the map of the pixels classified as land [orange] and water [blue] used in (a) the GOES-R ABI data and in (b) the KHGX WSR-88D data. The locations of the sites used are shown with black symbols.**




**Figure 2: On 23 July 2022, (a) time-height mapping of KHGX WSR-88D radar reflectivity at all elevation angles in the 0.5 km x 0.5 km grid point over the TRACER AFM1 site; (b) time-height mapping of radar reflectivity [color] and cloud boundaries [black points] from the ARSCL product at the TRACER AMF1; and (c) time-height mapping of GOES-R ABI Channel 13 brightness temperatures [black] and GOES-R ABI Channel 2 reflectance [red] in the 0.5 km**



**x 0.5 km grid point over the TRACER AMF1. Gray shading indicates the periods when the GOES-R ABI cloud mask**

**has values of 'cloudy'.**

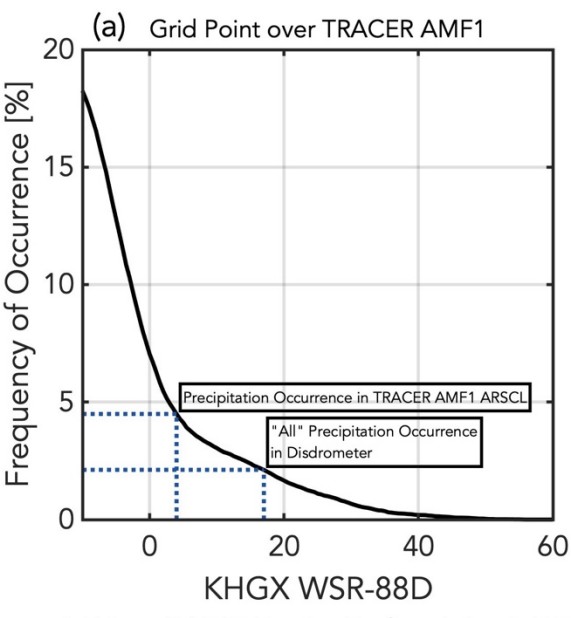

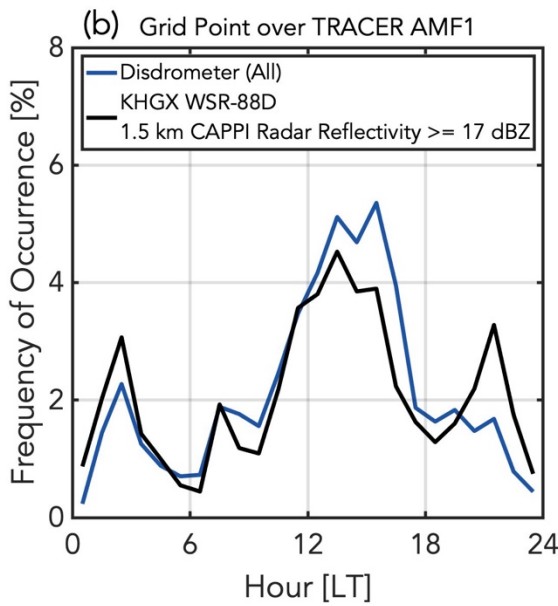

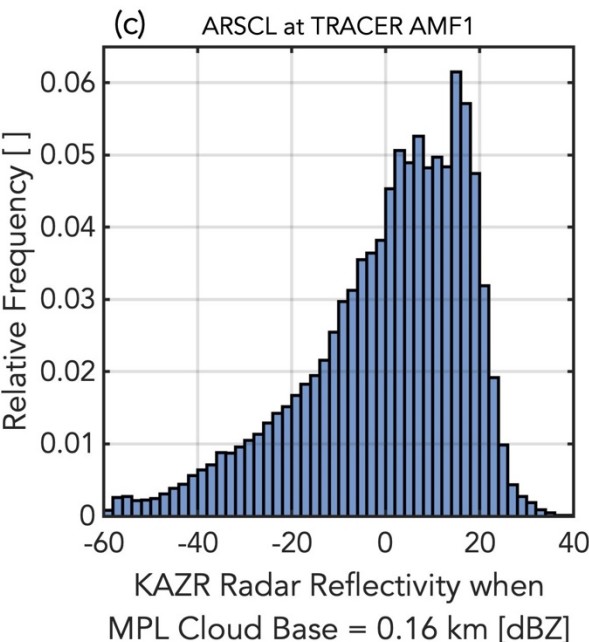

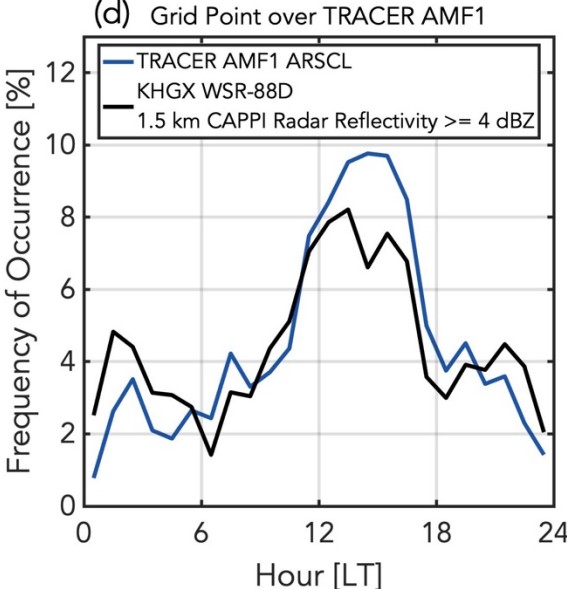

**Figure 3: From 1 June 2022 to 30 September 2022, (a) the frequency of occurrence of cumulative radar reflectivity**

**thresholds in the KHGX WSR-88D 1.5 km CAPPI radar reflectivity data for the 0.5 km x 0.5 km grid point over the**

**TRACER AMF1; (b) the frequency of occurrence every hour of all non-drizzle precipitation from the disdrometer at**



the TRACER AMF1 [blue] and of radar reflectivity values greater than or equal to 17 dBZ in the KHGX WSR-88D 1.5 km CAPPI radar reflectivity data for the 0.5 km x 0.5 km grid point over the TRACER AMF1 [black]; (c) histogram of radar reflectivity values from the TRACER AMF1 ARSCL product when there is a cloud base height recorded in the first range gate (0.16 km); and (d) the frequency of occurrence every hour of TRACER AMF1 ARSCL cloud base
heights of 0.16 km [blue] and of radar reflectivity values greater than or equal to 4 dBZ in the KHGX WSR-88D 1.5 km CAPPI radar reflectivity data for the 0.5 km x 0.5 km grid point over the TRACER AMF1 [black].

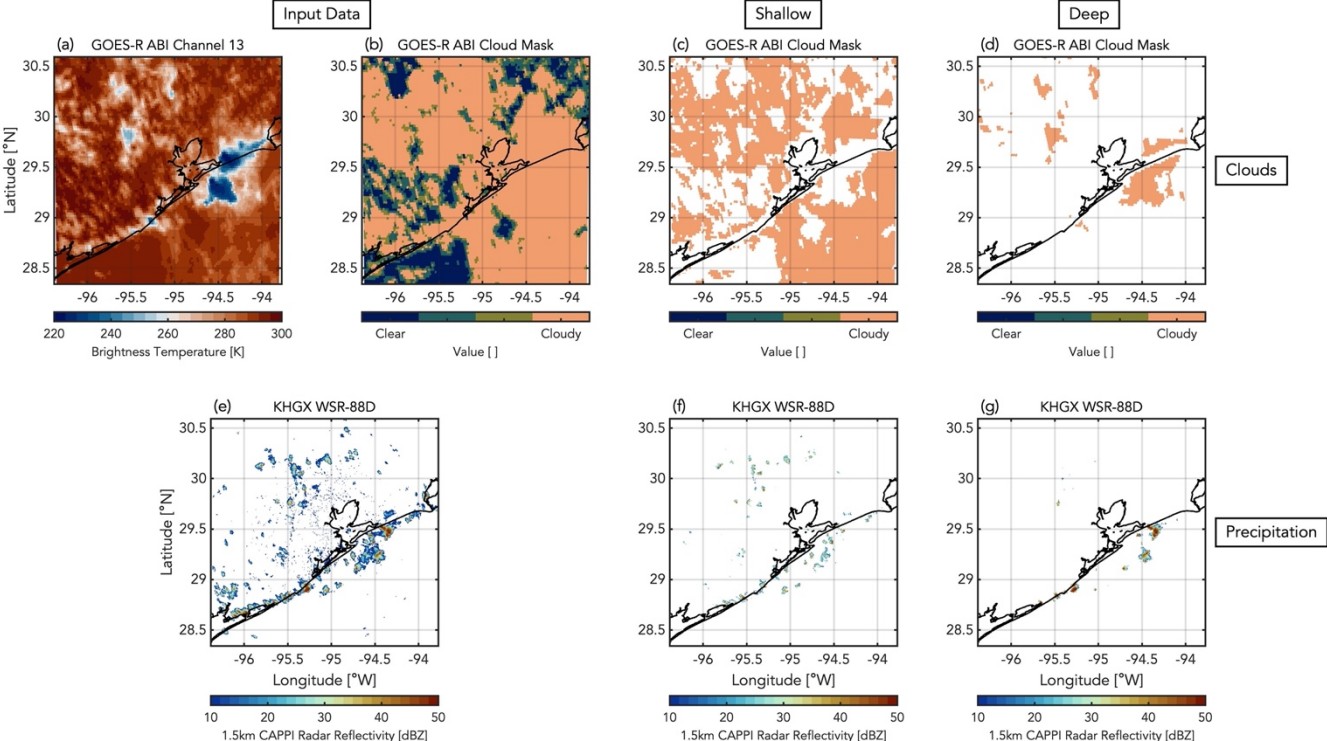

Figure 4: On 2 June 2022 at 11:22:36 LT, (a) GOES-R ABI Channel 13 brightness temperatures; (b) GOES-R ABI Cloud Mask values; (c) GOES-R ABI Cloud Mask warm phase cloudy pixels; (d) GOES-R ABI Cloud Mask mixed
phase cloudy pixels; (f) KHGX WSR-88D 1.5 km CAPPI radar reflectivity values; (g) KHGX WSR-88D 1.5 km CAPPI radar reflectivity values associated with warm phase echo tops; and (h) KHGX WSR-88D 1.5 km CAPPI radar reflectivity values associated with cold phase echo tops.





**Figure 5: From 1 June 2022 to 30 September 2022, the daily diurnal cycle in the Houston domain of (a) median shallow cloud fraction and (b) median deep cloud fraction. White spaces indicate missing data.**



**Figure 6: From 1 June 2022 to 30 September 2022, the daily diurnal cycle in the Houston domain of (a) median shallow precipitation fraction and (b) median deep precipitation fraction. White spaces indicate missing data.**



Figure 7: From 1 June 2022 to 30 September 2022, the four modes of diurnal shallow cloud fraction in the Houston domain using k-means clustering. 48 days belong in Cluster 1 (a), 21 days belong in Cluster 2 (b), 22 days belong in Cluster 3 (c), and 30 days belong in Cluster 4 (d). Lines indicate the hourly median, and shading indicates the hourly interquartile range.



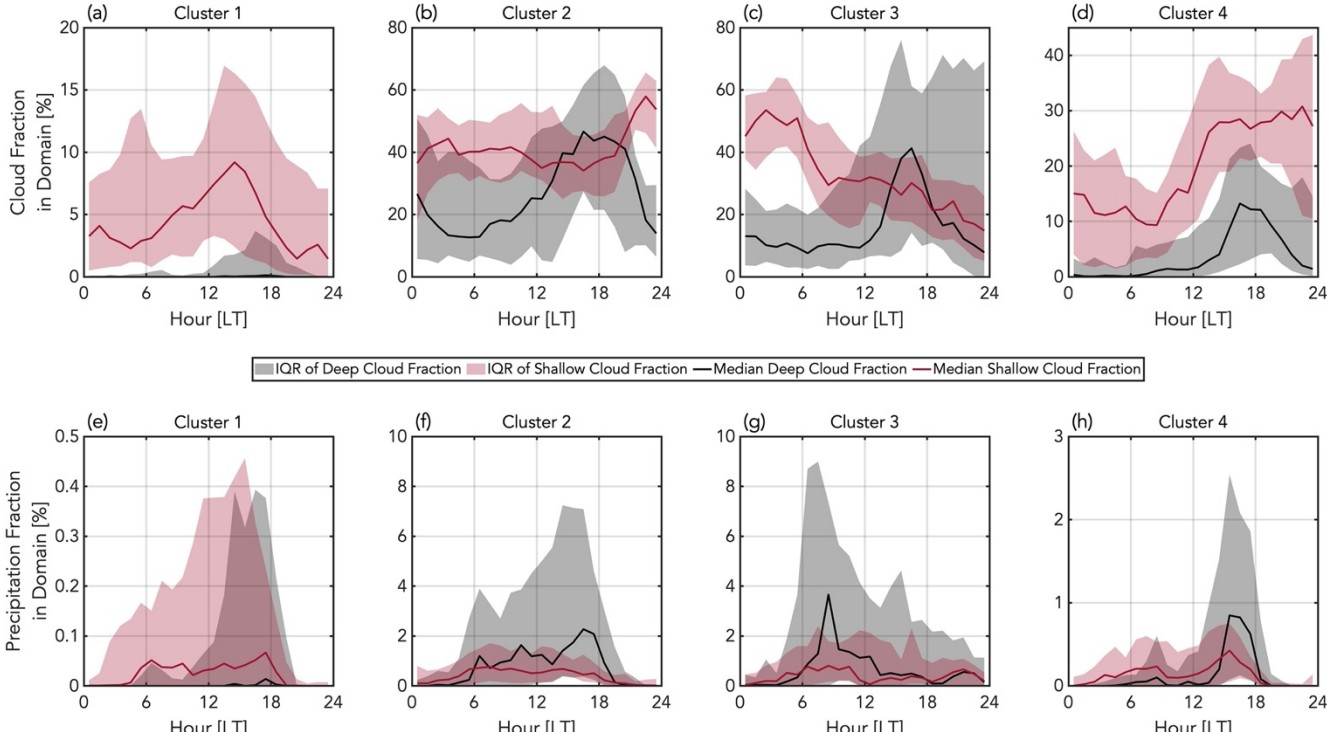

**Figure 8: The diurnal shallow [red] and deep [black] cloud (a-d) and precipitation (e-h) fractions in the Houston domain corresponding to the four k-means clusters of diurnal shallow cloud fraction. Lines indicate the hourly median, and shading indicates the hourly interquartile range.**



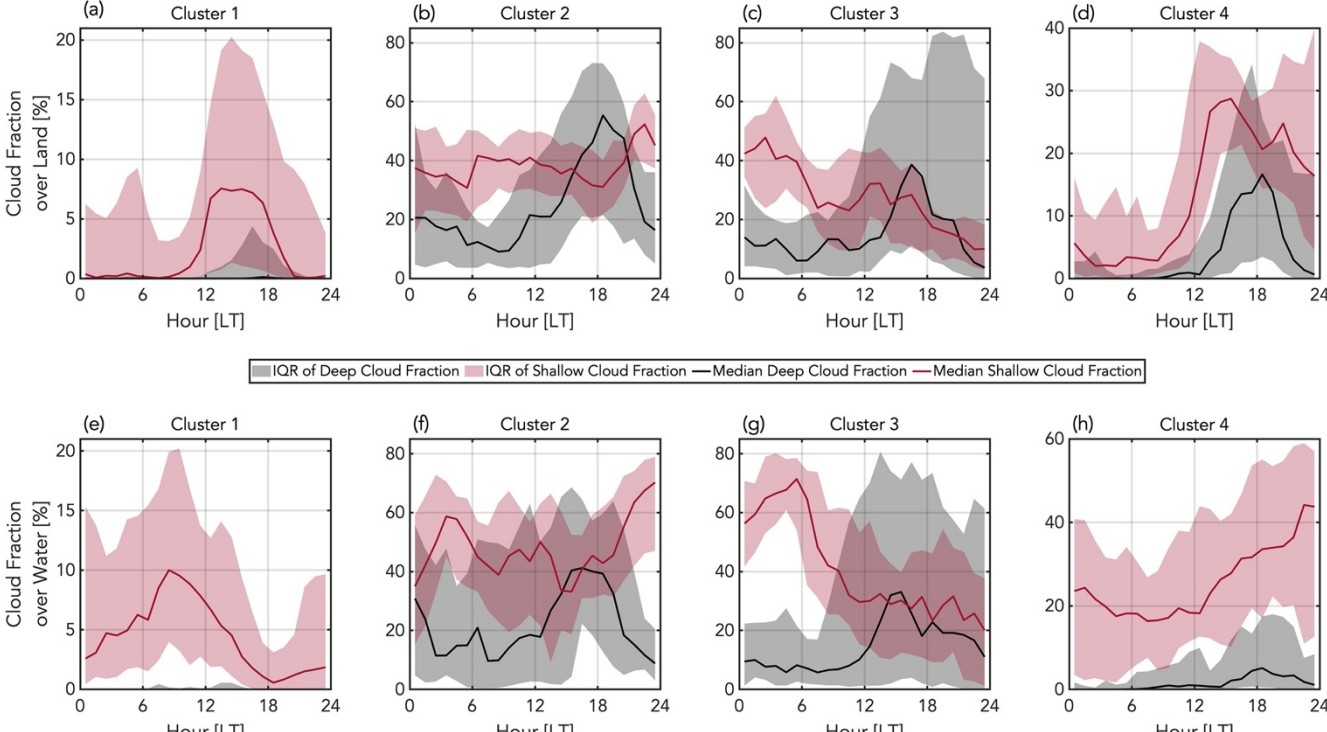

**Figure 9: The diurnal shallow [red] and deep [black] cloud fractions over land (a-d) and over water (e-h) corresponding to the four k-means clusters of domain-wide diurnal shallow cloud fraction. Lines indicate the hourly median, and shading indicates the hourly interquartile range.**




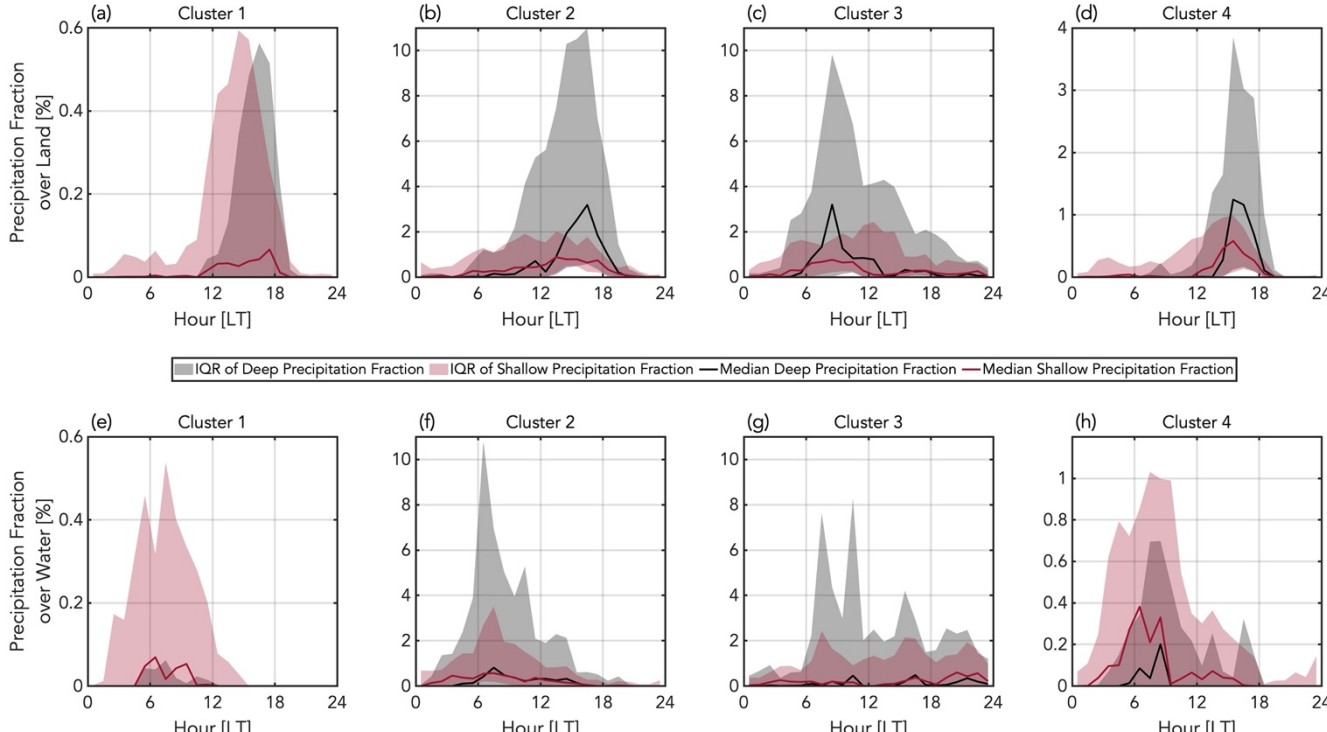

**Figure 10: The diurnal shallow [red] and deep [black] precipitation fractions over land (a-d) and over water (e-h) corresponding to the four k-means clusters of domain-wide diurnal shallow cloud fraction. Lines indicate the hourly median, and shading indicates the hourly interquartile range.**







**Figure 11: The diurnal cycle of the percentage of clouds [solid] and precipitation [dashed] classified as shallow over land (a-b) and over water (c-d) in Clusters 1 and 4.**





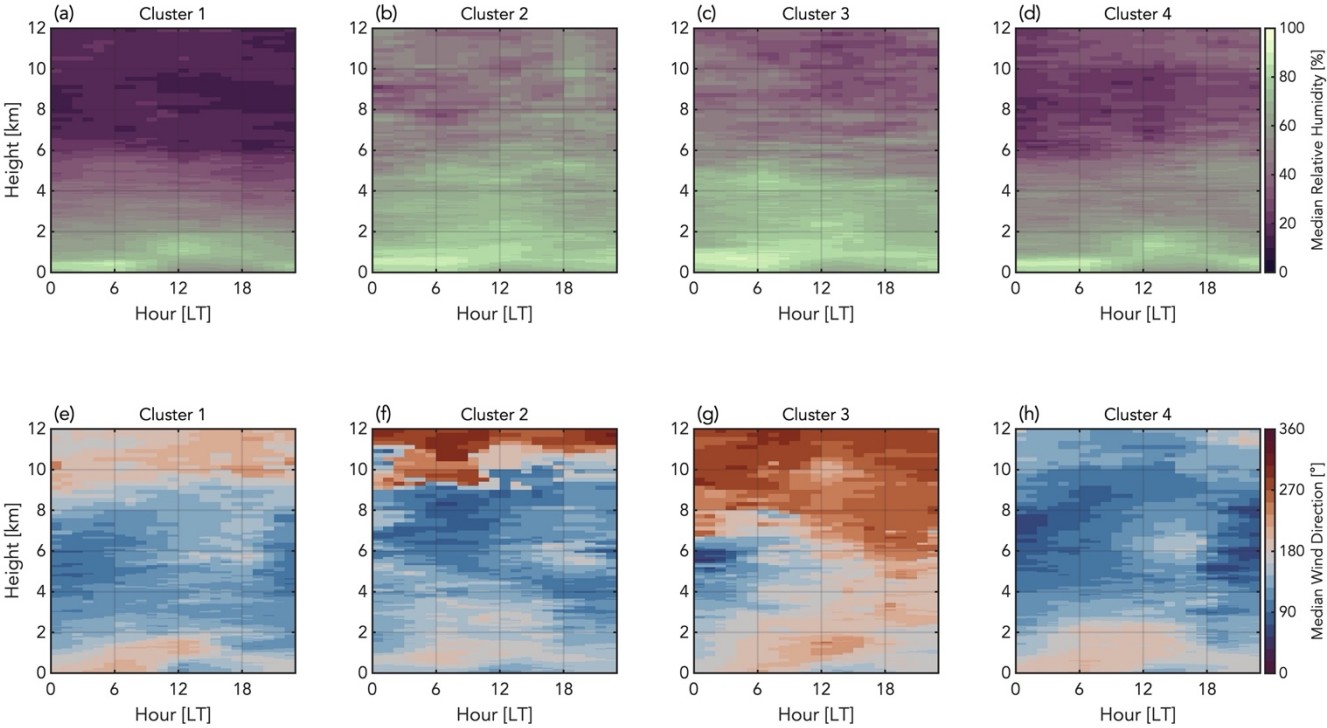

**Figure 12: Time-height mapping of the median relative humidity (a-d) and median wind direction (e-h) from the INTERPSONDE value-added product at the AMF1 corresponding to the four k-means clusters of shallow cloud fraction.**

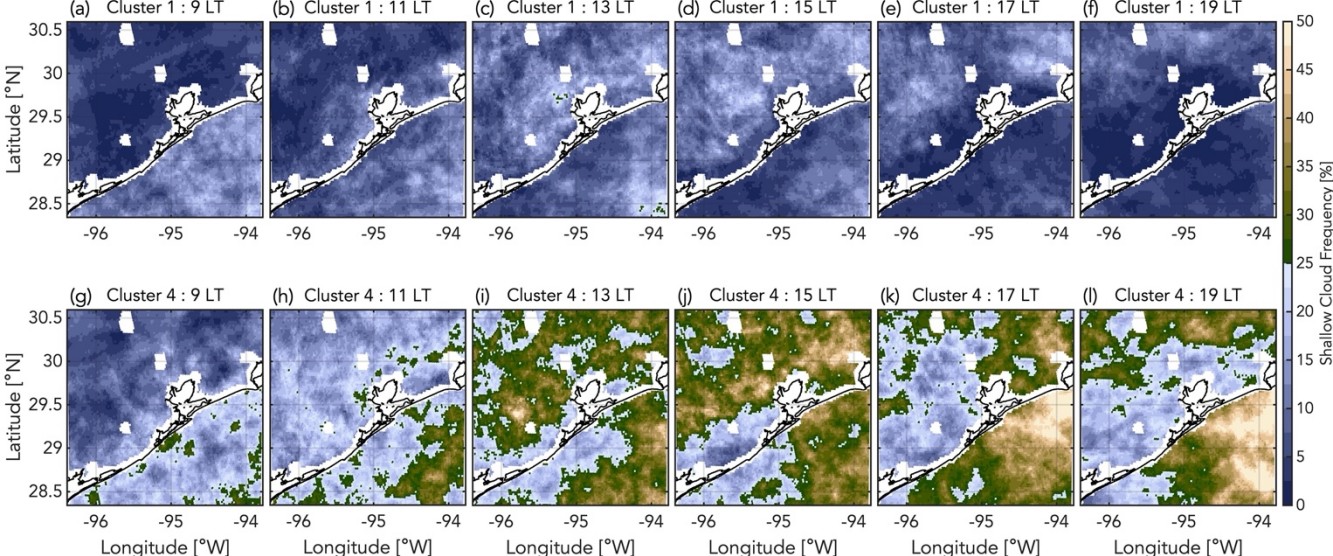

**Figure 13: The frequency of shallow cloud occurrences at every pixel every two hours from 9:00 to 19:00 LT for Cluster 1 days (top row) and Cluster 4 days (bottom row).**



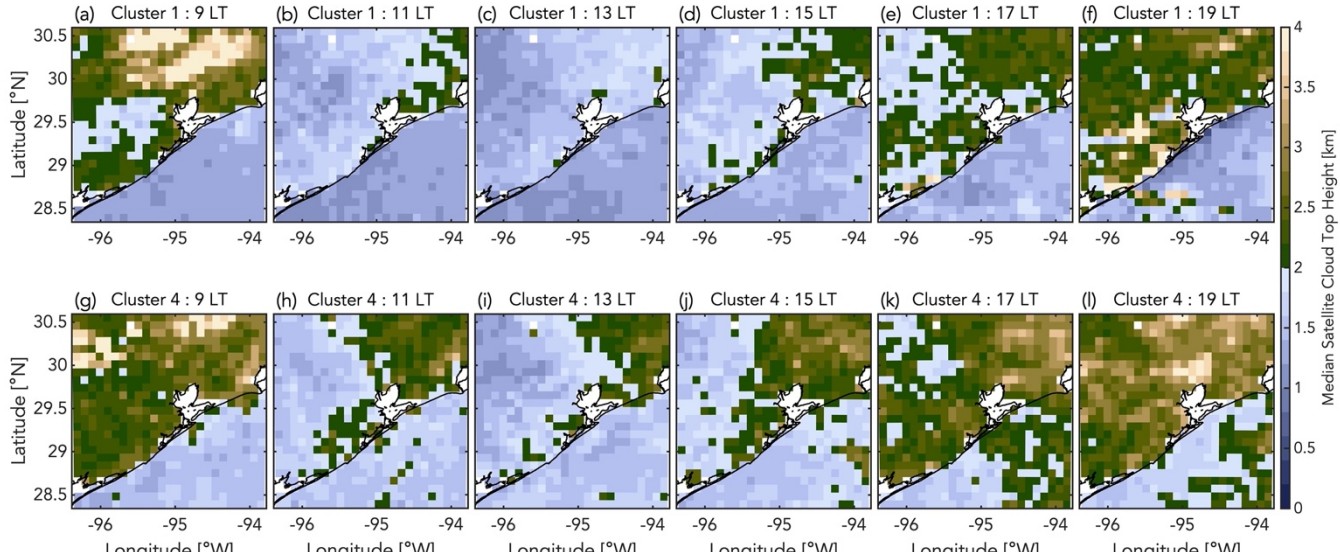

**Figure 14: The median cloud top height of shallow clouds in each 10 km x 10 km box every two hours from 9:00 to 19:00 LT for Cluster 1 days (top row) and Cluster 4 days (bottom row).**

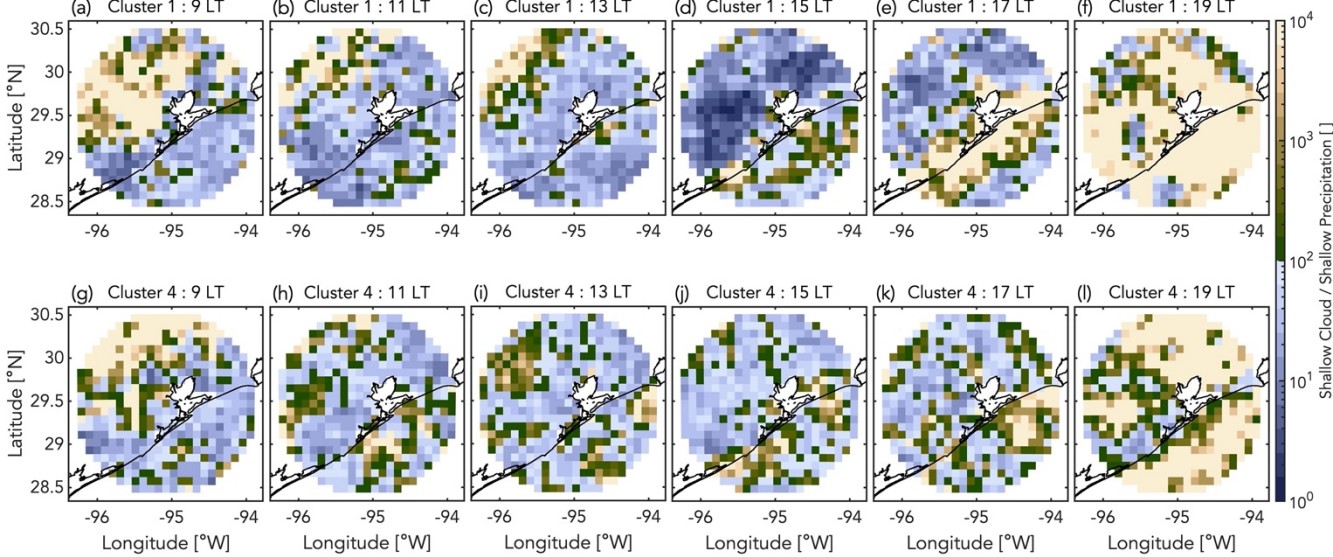

995    **Figure 15: The ratio of shallow cloud occurrences to shallow precipitating cloud occurrences in each 10 km x 10 km box every two hours from 9:00 to 19:00 LT for Cluster 1 days (top row) and Cluster 4 days (bottom row).**





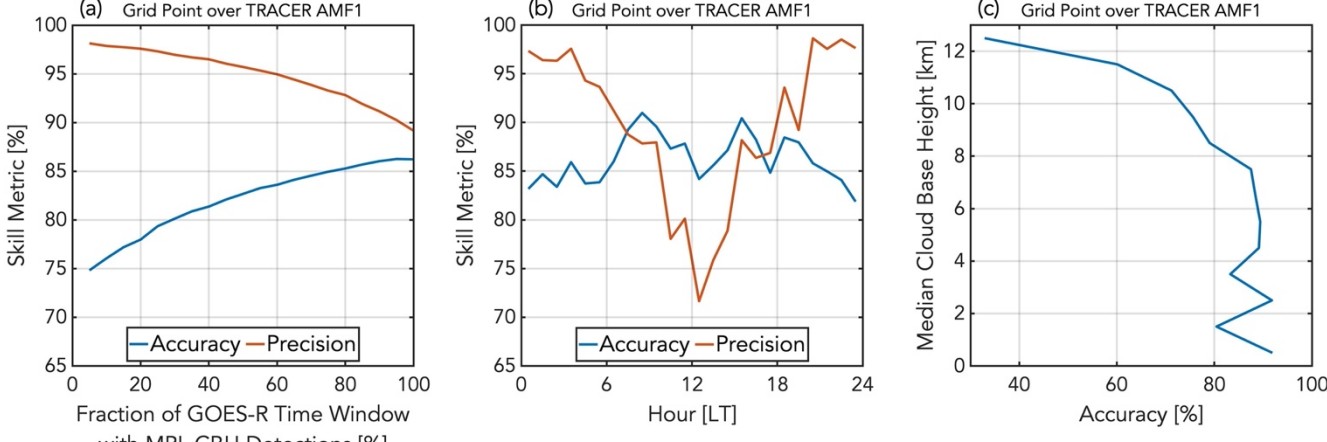

**Figure A1: From 1 June 2022 to 30 September 2022, using the cloud base height measurements from the TRACER AMF1 ARSCL product as the truth and the GOES-R ABI cloud mask values in the 0.5 km x 0.5 km grid point over the TRACER AMF1 as the prediction, the accuracy [blue] and precision [orange] of the GOES-R ABI cloud mask as a function of (a) the percentage of the GOES-R time window containing ARSCL cloud base height detections in order to for the "truth" to be designated as cloudy and (b) as a function of the local hour; and (c) the accuracy of the GOES-R ABI cloud mask for increasing 1 km-spaced bins of ARSCL median cloud base height during each five-minute GOES-R time window.**

1000





**Figure B1: Monthly calendars for June-September 2022 denoting the four modes of diurnal shallow cloud fraction in the Houston domain each day is classified as. The modes are determined using k-means clustering.**

