# Peer review of "Shallow cloud variability in Houston, Texas during the ESCAPE and TRACER field experiments"

_EGUsphere, 2024_

## Referee Comment (RC1)

**Review of manuscript egusphere-2024-2984**

**Title:** Shallow cloud variability in Houston, Texas during the ESCAPE and TRACER field experiments

**Authors:** Zackary Mages, Pavlos Kollias, Bernat Puigdomènech Treserras, Paloma Borque, and Mariko Oue

**Summary:**

The authors use a multi-sensor data fusion approach to analyze the partition the cloud coverage and precipitation fraction from shallow and deep convective clouds observed over the Houston region during the TRACER and ESCAPE field campaigns between June and September 2022. The authors adopt a novel and creative methodology to differentiate between shallow and deep convective clouds while making use of NEXRAD and KAZR reflectivity data to estimate the diurnal variability in spatial coverage, cloud top height, and precipitation fraction of shallow and deep convective clouds. Using a clustering method, the authors identify four dominant modes of shallow convection wherein each cluster contains far more non-precipitating clouds, with a larger number of these nonprecipitating shallow clouds occurring over water. The manuscript is well-organized, and the results are generally presented coherently. However, the results and discussion sections rely heavily on exact percentages, which could disrupt the reading flow. The authors omit certain technical details and inherent assumptions of the scientific methods described in the methodology section that could be clarified in a revised version of the manuscript. A major assumption in this study is that meteorological characteristics—such as relative humidity, wind speed and direction, and cloud top temperature at AMF1—are inherently mapped to shallow and deep convective clouds across the entire 250 km x 250 km domain. This "critical" assumption should be explicitly acknowledged in the methodology and addressed in the discussion of the results, especially when other studies from the TRACER field campaign have explicitly quantified the mesoscale environmental variability across the sea breezes. Otherwise, the manuscript is in a good shape and within the scope of ACP, except for a few sections where more details or clarifications can be added to improve the quality of the manuscript. My recommendation is for major revision, and I encourage the authors to consider my feedback to address the concerns in the comments provided below:

**Major comments:**

1. **Lines 50-52**: This discussion is completely irrelevant for the purposes of this study. Please consider removing this.
2. **Lines 54-67:** The authors seem to overlook a key point regarding Houston's suitability as a site for the TRACER and ESCAPE field campaigns. This discussion becomes muddled with unnecessary details on mechanisms affecting cloud properties in the following paragraph since none of those are the focus of this study. I suggest reorganizing the paragraph to highlight the main points outlined in Fridlind et al. (2019), emphasizing Houston's unique advantages in the context of a relatively strong aerosol perturbation driven by mesoscale shallow circulations from the Gulf of Mexico sea breeze and Houston-Galveston Bay breeze. Additionally, the summer-time 500-hPa ridge pattern

leads to synoptically weak large-scale forcing that increases the likelihood of a diurnal convective cycle associated with the onshore flow. You may also want to refer other studies relevant to the TRACER field campaign, such as Wang et al. (2024) and Sharma et al. (2024).

**References:**

1. Fridlind, Ann M., et al. "Use of polarimetric radar measurements to constrain simulated convective cell evolution: A pilot study with Lagrangian tracking." Atmospheric Measurement Techniques 12.6 (2019): 2979-3000.
2. Wang, Dié, et al. "TRACER Perspectives on Gulf-Breeze and Bay-Breeze Circulations and Coastal Convection." *Monthly Weather Review* 152.10 (2024): 2207-2228.
3. Sharma, Milind, et al. "Observed Variability in Convective Cell Characteristics and Near-Storm Environments across the Sea-and Bay-Breeze Fronts in Southeast Texas." *Monthly Weather Review* 152.11 (2024): 2419-2441.

3. **Lines 69-87:** I recommend removing the discussion on aerosols, convective initiation, rainfall anomalies, urban dynamics, and the urban heat island effect, as these topics are outside the study's primary focus. Instead, consolidate this paragraph by integrating the relevant discussion on sea breeze circulations and shallow convection with the previous paragraph.
4. **Lines 101-103:** The italicized text summarizing the main objective of this study is a bit misleading. The authors do not characterize the control of aerosols (and to some extent even meteorology) in this study. The only meteorological variables analyzed include the relative humidity, and wind speed and direction. There is no discussion on convective instability (CAPE), inhibition (CIN), large-scale vertical velocity, among others, e.g., refer Marquis et al. (2023). I suggest rephrasing this sentence to make it more aligned with the main goals of this study.

**References:**
1. Marquis, James N., et al. "Near-cloud atmospheric ingredients for deep convection initiation." *Monthly Weather Review* 151.5 (2023): 1247-1267.

5. **Lines 109-110:** I am not quite sure what the authors mean by 'Days with deep clouds are identified and ignored…' Aren't Figs. 4,5,6,8,9, and 10 comparing the deep convective cloud activity over the domain? Additionally, do the authors not account for precipitation from deep convection to arrive at the frequency of occurrence of precipitation at AMF1(2.2%)? Please clarify or revise as needed.

6. **Lines 123-128:** If the variables mentioned here were primarily used as part of the INTERPSONDE VAP, then please revise and cite the correct product documentation reference.

7. **Lines 130-135:** Please mention the gate spacing and temporal resolution of KAZR radar reflectivity data.

8. **Lines 141-154:** Please mention the gate spacing, VCP mode, and average temporal resolution of the KHGX observations.

9. **Lines 150-154:** Sudden transition to 'three-dimensional gridded KHGX radar data' and without prior context and lack of details regarding the quality control steps. Additionally, the purpose of two different types of radar data, i.e., polar coordinate (native) and gridded radar data is not specified. Please revise.

10. **Lines 155-170:** Please include the temporal resolution of GOES-R ABI data products.

11. **Line 178:** Please include the spatial extent of the analysis domain in terms of the latitude and longitude of the lower left and upper right corners of the encompassing box.

12. **Lines 180-181:** How are categorical data such ACM classification of cloudy or non-cloudy pixel gridded onto a 500 m grid?

13. **Line 223:** Which precipitation type categories were included in this calculation? Both drizzle and raindrops?

14. **Line 227:** I am assuming that the KHGX CAPPI was chosen to be at 1.5 km ARL to ensure greater spatial coverage, but using the same argument as provided in lines 239-242, reflectivity at 1.5 km could be an inflated when some droplets evaporate between surface and 1.5 km. I am wondering how this might affect the KHGX reflectivity threshold and subsequent analysis.

15. **Line 228:** Why were the reflectivity values confined to the -10 to 60 dBZ range? Additionally, line 262 mentions that the 10 dBZ threshold still contains a considerable amount of nonmeteorological scatterers. Why use the -10 to 10 dBZ range then? Why not, just 10 to 60 dBZ range?

16. **Lines 245-247:** I am not sure why the histogram for KAZR reflectivity values matters for the analysis. Was it being used to inform the KHGX reflectivity threshold?

17. **Lines 271-287:** It is not obvious to me how the authors avoided mislabeling or misclassification of shallow and deep clouds using the cloud mask and cloud-top temperature products. In a hypothetical scenario, a cloudy pixel may be assigned to a cirrus cloud which is high enough in the troposphere to meet the cloud-top temperature threshold of < -5 deg C. Will that be classified as a deep cloud then? A similar misclassification can be imagined for a shallow cloud case. How were such cases avoided during this analysis. This information needs to be included in the methodology section.

18. **Lines 280-282:** Since the analysis domain is 250 km x 250 km, I am wondering how representative is the AMF1 sounding for clouds hundreds of kilometers away from the AMF1 site.

19. **Line 339:** Which high-resolution data are being talked about here? Similarly, what does all data in line 344 mean?

20. **Lines 356-359:** I disagree that cluster 1 and 4 have a similar evolution of shallow cloud fraction. While cluster 1 has a steady growth through early through and peaks around mid-afternoon before it declines, cluster 4 has a much more rapid growth right before the peak mid-afternoon and then becomes steady for the rest of the day.

21. **Lines 409-410:** It is difficult to figure out the peaks in Fig. 10g (over water) since the absolute magnitude is very small and there are multiple peaks.

22. **Line 421:** I would argue that the subsequent analysis presented in this section is not adequate to comment on what controls the shallow-to-deep transition. Please consider removing or rephrasing this sentence.

23. **Lines 435-437:** This sentence sounds quite vague, especially because there is no supporting analysis to back it up. Please consider removing or rephrasing it.

24. **Lines 439-461:** As pointed out in the summary above, assuming uniform meteorological conditions for convective clouds across the entire domain (and all four clusters) is problematic. Studies by Wang et al. (2024) and Sharma et al. (2024) demonstrate significant mesoscale variability around Houston—variability that motivated the TRACER and ESCAPE field campaigns. Additionally, the authors base this study's motivation on Houston's role as a natural laboratory for examining environmental and aerosol perturbation effects on shallow cloud evolution. I recommend incorporating additional TRACER radiosonde and profiling observations to assess meteorological variability across the four clusters.

25. **Lines 460-461:** As with the effects of humidity profile discussed above, can the authors offer some physically reasonable explanation for the wind effects on the characteristic differences across the clusters?

26. **Line 477:** Can the authors support their statement regarding 'stable marine boundary layer' using observed profiling/radiosonde data?

27. **Lines 491-492:** Same comment as above. Is the boundary layer stability a guess or a finding supported by observations?

28. **Line 500:** Why was the analysis changed from pixel-level data to a 10 km x 10 km box?

29. **Lines 528-532:** This discussion is irrelevant to the analysis, results, and prior discussion. Please consider removing it to avoid potential confusion.

30. **Lines 546-547:** This motivation should have been provided while introducing the cluster analysis in the results section as well.

**31. Lines 549-550:** Same comment as comment 29 above. This sentence is vague since none of the analysis presented in this study can help isolate the role of sea breeze circulation on shallow cloud properties. Please consider removing or rephrasing it.

**32. Lines 553-554:** How are the authors sure that stronger updrafts are driven by surface and aerosol properties? Is it possible that the terrain is higher in that region? Is it possible that the clouds over that region were part of a synoptic-scale system over the Gulf of Mexico that were often advected over land during the day?

**33. Lines 560-567:** Is this paragraph meant to serve as a direction for future work? If not, it sounds quite vague. Either way, I suggest rephrasing it to make the intention clear or remove it altogether.

**34. Lines 643-645:** It would be helpful to add URLs to the specific GOES-R ABI products used in this study. Also, are the authors planning to share the quality controlled KHGX data generated during MAAS cell tracking?

**Minor comments:**

1. **Line 94:** favors the subsequent development…

2. **Line 132:** Suggest replacing 'radar Doppler moments' with 'moment of the Doppler spectrum.'

3. **Line 137:** Suggest replacing 'counteracts' with 'complements.'

4. **Line 149:** This sentence has insufficient details about which 'two mechanically scanning radars' are being talked about.

5. **Line 151:** 'MAAS activities' sounds odd. Please revise to something like 'MAAS cell tracking analysis' or something similar.

6. **Line 164:** Is IGFOV different from IFOV. Please expand or revise this acronym.

7. **Lines 166-167:** Please specify the purpose for using reflectance and brightness temperature data here as well.

8. **Line 192:** Does that mean that the KHGX analysis region is a circle with diameter 225 km?

9. **Line 203:** Please use a consistent terminology to describe the data products. Either use 'cloud boundaries' or 'hydrometeor layer boundaries' if both these terms are being used interchangeably. If not, then please define them separately.

10. **Line 207:** Suggest using a specific time range instead of 'Early on…'

11. **Line 213:** May want to specify that two radars being discussed are KHGX and KAZR.

12. **Line 215:** Suggest rephrasing '…usage as surveillance' as '…usage for surveillance.'

13. **Line 217:** '…that can be used…'

14. **Line 245:** '… KAZR radar reflectivity values at 160 m AGL…'

15. **Line 255:** Please specify the furthest range.

16. **Line 260:** Please cite the relevant source for the Z-R relationship.

17. **Line 281:** Please specify the reflectivity threshold used to define the echo-top height.

18. **Lines 373-376:** Suggest moving this sentence to line 372 before 'C1 and C4…' to maintain continuity of discussion regarding clusters 2 and 3.

19. **Line 385:** 'C4  has this feature…'

20. **Line 420:** 'Given these two clusters feature…'

21. **Line 445:** What does 'these afternoon hours' refer to?

22. **Line 518:** Typo error: 'figure 11d' should be 'figure 13d' instead.

23. **Lines 521-522:** This is a standalone sentence with no prior context. Please add more context or consider removing it.

24. **Line 548:** Suggest replacing 'healthy' with 'robust' or something similar.

25. **Line 614:** Please specify the highest precision value.

26. **Line 940:** '…grid point over  TRACER AMF1.'

**Figures:**

1. **Figures 1,4,13,14,15:** A negative sign in front of longitude values indicates locations east of the Prime Meridian. I suggest either removing the negative sign from longitude values or omitting [°W] from the x-axis labels to avoid redundancy.

2. **Figure 1:** Suggest specifying that the KHGX pixels are confined to a 112.5 km range for the KHGX radar in the figure caption. The locations/sites depicted through markers should also be specified either directly in the plots or specified in the caption.

3. **Figure 2:** It may be helpful to add information regarding the set of elevation angles used in panel plot (a). Suggest replacing 'radar' with 'KAZR' in line 937.

4. **Figure 4:** The reflectivity data in panel plot (f) shows some convective cells in the region close to the coast which was masked out for analysis. Can the authors clarify what's going on here?

5. **Figure A1:** In the panel plot (a), does MPL CBH mean the same thing as AMF1 ARSCL product? If so, then use one term to be consistent.

---

## Author Comment (AC1)

Review of manuscript egusphere-2024-2984

Title: Shallow cloud variability in Houston, Texas during the ESCAPE and TRACER field experiments

Authors: Zackary Mages, Pavlos Kollias, Bernat Puigdomènech Treserras, Paloma Borque, and Mariko Oue

Summary:
The authors use a multi-sensor data fusion approach to analyze the partition the cloud coverage and precipitation fraction from shallow and deep convective clouds observed over the Houston region during the TRACER and ESCAPE field campaigns between June and September 2022. The authors adopt a novel and creative methodology to differentiate between shallow and deep convective clouds while making use of NEXRAD and KAZR reflectivity data to estimate the diurnal variability in spatial coverage, cloud top height, and precipitation fraction of shallow and deep convective clouds. Using a clustering method, the authors identify four dominant modes of shallow convection wherein each cluster contains far more non-precipitating clouds, with a larger number of these nonprecipitating shallow clouds occurring over water. The manuscript is well-organized, and the results are generally presented coherently. However, the results and discussion sections rely heavily on exact percentages, which could disrupt the reading flow. The authors omit certain technical details and inherent assumptions of the scientific methods described in the methodology section that could be clarified in a revised version of the manuscript. A major assumption in this study is that meteorological characteristics—such as relative humidity, wind speed and direction, and cloud top temperature at AMF1—are inherently mapped to shallow and deep convective clouds across the entire 250 km x 250 km domain. This "critical" assumption should be explicitly acknowledged in the methodology and addressed in the discussion of the results, especially when other studies from the TRACER field campaign have explicitly quantified the mesoscale environmental variability across the sea breezes. Otherwise, the manuscript is in a good shape and within the scope of ACP, except for a few sections where more details or clarifications can be added to improve the quality of the manuscript. My recommendation is for major revision, and I encourage the authors to consider my feedback to address the concerns in the comments provided below:

Thank you to Reviewer #1 for the valuable feedback. We believe the manuscript has been improved immensely because of it. In summary, to accommodate R1's concerns, we focused on reworking the introduction and the summary/discussion sections to provide better and clearer context for the current work our study fits in with. We provided more concise details about our datasets, and we sharpened our methodology and motivation for our results. Most importantly, we added a new figure which shows meteorological context from HRRR model output throughout the domain rather than one location's set of radiosondes.
* * *
**MAJOR COMMENTS**

**Major Comment 1**: (Lines 50-52) This discussion is completely irrelevant for the purposes of this study. Please consider removing this.

We removed this.

**Major Comment 2**: (Lines 54-67) The authors seem to overlook a key point regarding Houston's suitability as a site for the TRACER and ESCAPE field campaigns. This discussion becomes muddled with unnecessary details on mechanisms affecting cloud properties in the following paragraph since none of those are the focus of this study. I suggest reorganizing the paragraph to highlight the main points outlined in Fridlind et al. (2019), emphasizing Houston's unique advantages in the context of a relatively strong aerosol perturbation driven by mesoscale shallow circulations from the Gulf of Mexico sea breeze and Houston-Galveston Bay breeze. Additionally, the summer-time 500-hPa ridge pattern leads to synoptically weak large-scale forcing that increases the likelihood of a diurnal convective cycle associated with the onshore flow. You may also want to refer other studies relevant to the TRACER field campaign, such as Wang et al. (2024) and Sharma et al. (2024).

> References:
> Fridlind, Ann M., et al. "Use of polarimetric radar measurements to constrain simulated convective cell evolution: A pilot study with Lagrangian tracking." Atmospheric Measurement Techniques 12.6 (2019): 2979-3000.
>
> Wang, Dié, et al. "TRACER Perspectives on Gulf-Breeze and Bay-Breeze Circulations and Coastal Convection." Monthly Weather Review 152.10 (2024): 2207- 2228.
>
> Sharma, Milind, et al. "Observed Variability in Convective Cell Characteristics and Near-Storm Environments across the Sea-and Bay-Breeze Fronts in Southeast Texas." Monthly Weather Review 152.11 (2024): 2419-2441.

Thank you for this suggestion. The introduction needed some work. We have reorganized and consolidated the section, using the references you suggested.

**Major Comment 3**: (Lines 69-87) I recommend removing the discussion on aerosols, convective initiation, rainfall anomalies, urban dynamics, and the urban heat island effect, as these topics are outside the study's primary focus. Instead, consolidate this paragraph by integrating the relevant discussion on sea breeze circulations and shallow convection with the previous paragraph.

We removed most of this discussion and reorganized the introduction.

**Major Comment 4**: (Lines 101-103) The italicized text summarizing the main objective of this study is a bit misleading. The authors do not characterize the control of aerosols (and to some extent even meteorology) in this study. The only meteorological variables analyzed include the relative humidity, and wind speed and direction. There is no discussion on convective instability (CAPE), inhibition (CIN), large-scale vertical velocity, among others, e.g., refer Marquis et al. (2023). I suggest rephrasing this sentence to make it more aligned with the main goals of this study.

References:

Marquis, James N., et al. "Near-cloud atmospheric ingredients for deep convection initiation." Monthly Weather Review 151.5 (2023): 1247-1267.

We revised this section for clarity.

**Major Comment 5**: (Lines 109-110) I am not quite sure what the authors mean by 'Days with deep clouds are identified and ignored...' Aren't Figs. 4,5,6,8,9, and 10 comparing the deep convective cloud activity over the domain? Additionally, do the authors not account for precipitation from deep convection to arrive at the frequency of occurrence of precipitation at AMF1(2.2%)? Please clarify or revise as needed.

We revised this section for clarity.

**Major Comment 6**: (Lines 123-128) If the variables mentioned here were primarily used as part of the INTERPSONDE VAP, then please revise and cite the correct product documentation reference.

We are redoing our meteorological analysis using HRRR data instead of the TRACER INTERPSONDE data, so we are removing this dataset entirely.

**Major Comment 7**: (Lines 130-135) Please mention the gate spacing and temporal resolution of KAZR radar reflectivity data.

We added: "during TRACER, and it has a vertical resolution of 30m and a temporal resolution of 2 s."

**Major Comment 8**: (Lines 141-154) Please mention the gate spacing, VCP mode, and average temporal resolution of the KHGX observations.

We added two sentences to address this:
1) "These radars gave a range resolution of 250m, an azimuthal resolution of 1°, and a range of temporal resolutions from 4 to 10 minutes depending on the Volume Coverage Pattern (VCP)."
2) "KHGX operated in clear air mode, using VCPs 31, 32, and 35, and in precipitation mode, using VCPs 21, 212, and 215, in the four months of interest."

**Major Comment 9**: (Lines 150-154) Sudden transition to 'three-dimensional gridded KHGX radar data' and without prior context and lack of details regarding the quality control steps. Additionally, the purpose of two different types of radar data, i.e., polar coordinate (native) and gridded radar data is not specified. Please revise.

We reworked the paragraph to now read as:
"During the TRACER and ESCAPE IOPs, the KHGX WSR-88D observations were processed in real time to support the operation of the Multisensor Agile Adaptive Sampling (MAAS; Kollias et al., 2020a; Lamer et al., 2023) algorithm implanted in the second-generation C-band scanning ARM precipitation radar (CSAPR2; Kollias et al. 2020b) and the Colorado State University C-

band Hydrological Instrument for Volumetric Observation (CHIVO) radar to conduct convective cell tracking (Lamer et al., 2023; Kollias et al., 2024). The three-dimensional KHGX data (two dimensions in azimuth and one dimension in elevation) were first quality-controlled for nonmeteorological echoes and then interpolated onto a horizontal Cartesian grid for comparison with satellite data. These gridded and masked data were then used to construct a 1.5 km above ground level (AGL) constant-altitude plan position indicator (CAPPI; Douglas, 1990) and a map of vertically-integrated liquid (VIL) in the domain. VIL is estimated using the Marshall-Palmer drop size distribution (Marshall and Palmer, 1948) assumptions:

$$VIL = \sum_{i=0}^{top} 3.44 \times 10^{-6} [\frac{Z_i + Z_{i+1}}{2}]^{4/7} \Delta h$$

where Z stands for radar reflectivity (in dBZ) and $\Delta h$ stands for the depth of the layer between consecutive grid levels (in m). More specific details on these processes can be found in Lamer et al. (2023). For this study, we use the KHGX CAPPI, VIL, and three-dimensional data."

**Major Comment 10**: (Lines 155-170) Please include the temporal resolution of GOES-R ABI data products.

We made the following changes to address this:
1) "Channel 2 sits in the wavelength range of 0.59-0.69um, has an instantaneous field of view (IFOV) of 0.5km, and has a temporal resolution of 5 minutes…"
2) "Channel 13 sits in the wavelength range of 10.1-10.6um, has an IFOV of 2 km, and has a temporal resolution of 5 minutes…"
3) "The ABI cloud mask (hereafter ACM) product combines nine of the 16 bands to provide a binary classification for each pixel using spectral, special, and temporal signatures and has a horizontal resolution of 2 km and temporal resolution of 5 minutes…"

**Major Comment 11**: (Line 178) Please include the spatial extent of the analysis domain in terms of the latitude and longitude of the lower left and upper right corners of the encompassing box.

We added the following sentence: "In terms of spatial extent, the southwestern-point of the domain is 28 20' 35.8794" N, 96 21' 16.56" W, and the northeastern-most point of the domain I 30 35' 17.26" N, 93 46' 31.08" W."

**Major Comment 12**: (Lines 180-181) How are categorical data such ACM classification of cloudy or non- cloudy pixel gridded onto a 500 m grid?

We use nearest-neighbor interpolation.

**Major Comment 13**: (Line 223) Which precipitation type categories were included in this calculation? Both drizzle and raindrops?

We clarified this: "Measured surface precipitation, which includes drizzle and rain, is observed…"

**Major Comment 14**: (Line 227) I am assuming that the KHGX CAPPI was chosen to be at 1.5 km ARL to ensure greater spatial coverage, but using the same argument as provided in lines 239- 242, reflectivity at 1.5 km could be an inflated when some droplets evaporate between surface and 1.5 km. I am wondering how this might affect the KHGX reflectivity threshold and subsequent analysis.

Great point. We could have taken two approaches to establish our threshold. The first approach could have used a Z-R relationship for this regime, but this would have encountered the problem you suggested, where we would not be accounting for evaporation because we were tuning at 1.5 km. We instead tuned the threshold to precipitation that reaches the surface. Our threshold essentially means that radar reflectivities greater than 7.5 dBZ at 1.5 km are reaching the surface, and a disdrometer placed anywhere in the domain would see this relationship. Given the different wavelengths between KHGX and KAZR, we were struggling to get our relationships to match, so we eliminated that part of the methodology and chose the surface-constrained approach in the KHGX data.

**Major Comment 15**: (Line 228): Why were the reflectivity values confined to the -10 to 60 dBZ range? Additionally, line 262 mentions that the 10 dBZ threshold still contains a considerable amount of non-meteorological scatterers. Why use the -10 to 10 dBZ range then? Why not, just 10 to 60 dBZ range?

Great questions and points. This made us reassess this part of the methodology. Figure 2 clearly shows that nonmeteorological echoes (biological activity, clutter) made it through the extensive quality control done on the dataset by Lamer et al. (2023). So, we decided to apply the VIL threshold first and then examine the radar reflectivity values present over the site that made it through this filtering. We only applied the VIL threshold after choosing the threshold in the first version. The range of radar reflectivities that occur is from -10 to 60 dBZ (same as before), and we match the frequency of occurrence to the frequency of occurrence of surface measured precipitation measured by the AMF1 disdrometer. Our threshold decreased slightly to 7.5 dBZ. This is still above the minimum detectable signal measured by KHGX at far ranges, so we are clear to use this one. We removed the ARSCL considerations entirely.

**Major Comment 16**: (Lines 245-247) I am not sure why the histogram for KAZR reflectivity values matters for the analysis. Was it being used to inform the KHGX reflectivity threshold?

We had originally included it for general interest and to highlight the differences between the two radar reflectivity datasets in precipitation conditions. It was not used to inform the KHGX reflectivity threshold.

However, we have subsequently removed the ARSCL component of this methodology, and Figure 3 does not have this panel anymore.

**Major Comment 17**: (Lines 271-287) It is not obvious to me how the authors avoided mislabeling or misclassification of shallow and deep clouds using the cloud mask and cloud-top temperature products. In a hypothetical scenario, a cloudy pixel may be assigned to a cirrus cloud which is high enough in the troposphere to meet the cloud-top temperature threshold of < -

5 deg C. Will that be classified as a deep cloud then? A similar misclassification can be imagined for a shallow cloud case. How were such cases avoided during this analysis. This information needs to be included in the methodology section.

This is a great point. We do not account for these ambiguous classifications. Anything from a cirrus cloud to an anvil to a convective cumulus counts as a deep cloud if it is the coldest/highest feature in the column. This also means that shallow clouds underneath these features may be obscured. This partially motivated why we focused on the prominent shallow cloud clusters (C1 and C4) because the more widespread deep cloud activity in C3 and C4 may be underrepresenting the statistics. This highlights a difficulty in this type of methodology, so we will make this clearer and add more clarity to our motivations.

**Major Comment 18**: (Lines 280-282) Since the analysis domain is 250 km x 250 km, I am wondering how representative is the AMF1 sounding for clouds hundreds of kilometers away from the AMF1 site.

We had a similar thought. However, we looked at some statistics surrounding this temperature level in the soundings, and we do not see much variation in the heights, even in the diurnal cycle in Panel B. We are observing the middle atmosphere, where any sort of disturbance most likely would've been seen in the observations. Given we are not looking at the more variable boundary layer for example, we are comfortable with applying one height/temperature across the domain, similarly to what we did for the satellite data.

[Figure]

**Major Comment 19**: (Line 339) Which high-resolution data are being talked about here? Similarly, what does all data in line 344 mean?

We made the following changes:
1) "is applied to our satellite-based shallow cloud data to identify…"
2) "Finally, for each cluster, we identify the applicable days and calculate…"

**Major Comment 20**: (Lines 356-359) I disagree that cluster 1 and 4 have a similar evolution of shallow cloud fraction. While cluster 1 has a steady growth through early through and peaks around mid-afternoon before it declines, cluster 4 has a much more rapid growth right before the peak mid-afternoon and then becomes steady for the rest of the day.

We accept your interpretation and will change accordingly:
"Finally, the 30 days in Cluster 4 (C4) are similar to C1 in that they both experience growth starting in the morning and continuing into the afternoon, but the growth rate is much steeper in C4. The shallow cloud fraction then becomes steady from the later afternoon and on, which does not occur in C1."

**Major Comment 21**: (Lines 409-410) It is difficult to figure out the peaks in Fig. 10g (over water) since the absolute magnitude is very small and there are multiple peaks.

We tweaked the y-axis limit to see the data more clearly while also trying to balance keeping all the data in view.

**Major Comment 22**: (Line 421) I would argue that the subsequent analysis presented in this section is not adequate to comment on what controls the shallow-to-deep transition. Please consider removing or rephrasing this sentence.

We removed this sentence.

**Major Comment 23**: (Lines 435-437) This sentence sounds quite vague, especially because there is no supporting analysis to back it up. Please consider removing or rephrasing it.

We removed this sentence.

**Major Comment 24**: (Lines 439-461) As pointed out in the summary above, assuming uniform meteorological conditions for convective clouds across the entire domain (and all four clusters) is problematic. Studies by Wang et al. (2024) and Sharma et al. (2024) demonstrate significant mesoscale variability around Houston—variability that motivated the TRACER and ESCAPE field campaigns. Additionally, the authors base this study's motivation on Houston's role as a natural laboratory for examining environmental and aerosol perturbation effects on shallow cloud evolution. I recommend incorporating additional TRACER radiosonde and profiling observations to assess meteorological variability across the four clusters.

This was a very valuable suggestion, and we believe the new figure we added addresses your concerns while providing value to the manuscript. We made a composite of HRRR 850hPa model output for the four clusters that provides broader spatial context for meteorological conditions and a view into the diurnal cycle at this level. We found drier conditions in the clusters with fewer deep clouds, and the relative humidity maxima at 850 hPa followed the highest frequencies of shallow clouds. The region had weak onshore flow in all clusters, and the temperature gradient was more aligned parallel with the coast all day in C1 and C4, which favors sea breeze conditions.

[Figure]

**Major Comment 25**: (Lines 460-461) As with the effects of humidity profile discussed above, can the authors offer some physically reasonable explanation for the wind effects on the characteristic differences across the clusters?

We removed the figure this comment was referencing.

**Major Comment 26**: (Line 477) Can the authors support their statement regarding 'stable marine boundary layer' using observed profiling/radiosonde data?

We reworded this to indicate we are hypothesizing.

**Major Comment 27**: (Lines 491-492) Same comment as above. Is the boundary layer stability a guess or a finding supported by observations?

We reworded this to indicate we are hypothesizing.

**Major Comment 28**: (Line 500) Why was the analysis changed from pixel-level data to a 10 km x 10 km box?

We do this to reduce some noisiness and enhance interpretability. We will make a note of that.

**Major Comment 29**: (Lines 528-532) This discussion is irrelevant to the analysis, results, and prior discussion. Please consider removing it to avoid potential confusion.

We removed it.

**Major Comment 30**: (Lines 546-547) This motivation should have been provided while introducing the cluster analysis in the results section as well.

We have now provided this motivation earlier in the paper.

**Major Comment 31**: (Lines 549-550) Same comment as comment 29 above. This sentence is vague since none of the analysis presented in this study can help isolate the role of sea breeze circulation on shallow cloud properties. Please consider removing or rephrasing it.

We reworded this section and placed it in the context of sea breeze work done in the region during this period.

**Major Comment 32**: (Lines 553-554) How are the authors sure that stronger updrafts are driven by surface and aerosol properties? Is it possible that the terrain is higher in that region? Is it possible that the clouds over that region were part of a synoptic-scale system over the Gulf of Mexico that were often advected over land during the day?

We are hypothesizing what this signal means, and you mention some possible explanations that we will add.

**Major Comment 33**: (Lines 560-567) Is this paragraph meant to serve as a direction for future work? If not, it sounds quite vague. Either way, I suggest rephrasing it to make the intention clear or remove it altogether.

We removed this section and rewrote the bulk of the discussion section.

**Major Comment 34**: (Lines 643-645) It would be helpful to add URLs to the specific GOES-R ABI products used in this study. Also, are the authors planning to share the quality controlled KHGX data generated during MAAS cell tracking?

We added links, and we made a note in the Data Availability section about our KHGX dataset.
* * *
**MINOR COMMENTS**

**Minor Comment 1**: (Line 94) favors the subsequent development...

We made this change.

**Minor Comment 2**: (Line 132) Suggest replacing 'radar Doppler moments' with 'moment of the Doppler spectrum.'

We made this change.

**Minor Comment 3**: (Line 137) Suggest replacing 'counteracts' with 'complements.'

We made this change.

**Minor Comment 4**: (Line 149) This sentence has insufficient details about which 'two mechanically scanning radars' are being talked about.

We added details and a citation for the two C-band mechanically-scanning radars we are referencing.

**Minor Comment 5**: (Line 151) 'MAAS activities' sounds odd. Please revise to something like 'MAAS cell tracking analysis' or something similar.

We changed from "MAAS activities" to "MAAS cell tracking output".

**Minor Comment 6**: (Line 164) Is IGFOV different from IFOV. Please expand or revise this acronym.

Thank you for catching a spelling mistake. "IGFOV" is supposed to be "IFOV" like in Line 162.

**Minor Comment 7**: (Lines 166-167) Please specify the purpose for using reflectance and brightness temperature data here as well.

We changed the sentence to read: "The reflectance data from Channel 2, which we use to help identify cloud, and the brightness temperature data from Channel 13, which we use to identify cloud phase, are used here."

**Minor Comment 8**: (Line 192) Does that mean that the KHGX analysis region is a circle with diameter 225 km?

Yes. We changed the phrase to "the region within the 225 km diameter range ring" for clarity.

**Minor Comment 9**: (Line 203) Please use a consistent terminology to describe the data products. Either use 'cloud boundaries' or 'hydrometeor layer boundaries' if both these terms are being used interchangeably. If not, then please define them separately.

We will stick with cloud boundaries. We made this change.

**Minor Comment 10**: (Line 207) Suggest using a specific time range instead of 'Early on...'

We added the phrase "Between 0 and 6 LT".

**Minor Comment 11**: (Line 213) May want to specify that two radars being discussed are KHGX and KAZR.

We tweaked the sentence to start as "The comparison between KAZR and KHGX also shows the value a WSR-88D…".

**Minor Comment 12**: (Line 215) Suggest rephrasing '...usage as surveillance' as '...usage for surveillance.'

We changed to "usage for operational surveillance".

**Minor Comment 13**: (Line 217) '...that can be used...'

We made this change.

**Minor Comment 14**: (Line 245) '... KAZR radar reflectivity values at 160 m AGL...'

We changed to "the corresponding histogram of KAZR radar reflectivity values at 160 m AGL".

**Minor Comment 15**: (Line 255) Please specify the furthest range.

We added: "At the furthest range considered in this study (~176 km from KHGX)".

**Minor Comment 16**: (Line 260) Please cite the relevant source for the Z-R relationship.

We added the citation.

**Minor Comment 17**: (Line 281) Please specify the reflectivity threshold used to define the echo-top height.

We changed to "comparing the highest KHGX echo top height".

**Minor Comment 18**: (Lines 373-376) Suggest moving this sentence to line 372 before 'C1 and C4...' to maintain continuity of discussion regarding clusters 2 and 3.

We made this change.

**Minor Comment 19**: (Line 385) 'C4  has this feature...'

We made this change.

**Minor Comment 20**: (Line 420) 'Given these two clusters feature...'

We made this change.

**Minor Comment 21**: (Line 445) What does 'these afternoon hours' refer to?

We changed for clarity: "C1 has the lowest relative humidity of the four clusters during the afternoon hours, …".

**Minor Comment 22**: (Line 518) Typo error: 'figure 11d' should be 'figure 13d' instead.

Thank you for catching this. We made the change.

**Minor Comment 23**: (Lines 521-522) This is a standalone sentence with no prior context. Please add more context or consider removing it.

We removed the sentence.

**Minor Comment 24**: (Line 548) Suggest replacing 'healthy' with 'robust' or something similar.

We changed from "healthy" to "robust".

**Minor Comment 25**: (Line 614) Please specify the highest precision value.

We tweaked the sentence to read: "…, and the precision is above 95% between 0-3 LT (local maximum of 97.6% at 3 LT) and 20-23 LT (absolute maximum of 98.6% at 20LT)".

**Minor Comment 26**: (Line 940) '...grid point over  TRACER AMF1.'

We made this change.
* * *
**FIGURES**

**Comment on Figures 1,4,13,14,15**: A negative sign in front of longitude values indicates locations east of the Prime Meridian. I suggest either removing the negative sign from longitude values or omitting [°W] from the x-axis labels to avoid redundancy.

Thank you for pointing this out; this is a detail that slipped by us. We adjusted the axes labels on all applicable figures.

**Comment on Figure 1**: Suggest specifying that the KHGX pixels are confined to a 112.5 km range for the KHGX radar in the figure caption. The locations/sites depicted through markers should also be specified either directly in the plots or specified in the caption.

We added some more details to the caption: "The pixels included in (b) are constrained to the region with the 225 km diameter range ring. The locations of TRACER AMF1, the KHGX WSR-88D, and downtown Houston are indicated by a square marker, a star marker, and a circle marker, respectively.

**Comment on Figure 2**: It may be helpful to add information regarding the set of elevation angles used in panel plot (a). Suggest replacing 'radar' with 'KAZR' in line 937.

We added "KAZR radar reflectivity", and we added the set of elevation angles, too.

**Comment on Figure 4**: The reflectivity data in panel plot (f) shows some convective cells in the region close to the coast which was masked out for analysis. Can the authors clarify what's going on here?

We believe you are referencing the convective cells south of 28.75ºN and west of 95.75ºW. These fall outside of the 225 km diameter range ring and are masked.

**Comment on Figure A1**: In the panel plot (a), does MPL CBH mean the same thing as AMF1 ARSCL product? If so, then use one term to be consistent.

Thank you for noticing this. Staying consistent is important. We changed to "ARSCL CBH detections [%]".

---

## Author Comment (AC2)

This study explores the variability of shallow convective clouds and precipitation in Houston, Texas, during the summer, utilizing data from the TRACER and ESCAPE field campaigns. Through a combination of geostationary satellite and ground-based radar observations over a 250x250 km domain, the authors identify four diurnal modes of shallow cloud fraction and analyze their spatial and temporal patterns. The study highlights the influence of land-ocean contrasts, sea breeze circulations, and local meteorology on shallow cloud properties, contributing to the understanding of cloud behavior in complex urban and coastal environments. Overall, while the study makes a contribution to understanding shallow cloud behavior in a complex urban and coastal environment, its impact is somewhat diminished by insufficient attention to physical mechanisms and scientific discussions. Addressing these issues would enhance the clarity and applicability of the findings.

Thank you to Reviewer #2 for the valuable feedback. We believe the manuscript has been improved immensely because of it. To accommodate R2's concerns, the main thing we did was provide a new figure which shows meteorological context from HRRR model output throughout the domain rather than one location's set of radiosondes.
* * *
**MAJOR COMMENTS**

**Major Comment 1**: The physical meaning of the identified regimes of shallow cumulus clouds is insufficiently addressed. While the clustering analysis identifies distinct modes, there is a lack of detailed discussion on their underlying drivers or implications for broader meteorological processes. It should explain more about the underlying differences in terms of the formation and mechanisms of different regimes.

In the new version of the manuscript, we expanded our discussion section to compare the prominent shallow cloud patterns we find to recent studies about synoptic patterns and sea breeze circulations in the region, and we provided a new figure with domain-wide meteorological context to make interpretations of our own. We wanted to provide some of our own insight while also acknowledging the many others who are looking at this region.

[Figure]

**Major Comment 2**: The variability of cloud fraction and precipitation is not thoroughly explained. Although the study presents statistical patterns, it does not adequately explore the mechanisms responsible for observed spatial and temporal differences, particularly in the context of different meteorological factors such as humidity, circulation, advection, etc.

We have added a new figure that shows composites of 850 hPa model output in each cluster, and we discuss the wind patterns, moisture, and temperature and its gradients across the domain as a function of time. We also connect our work to the recent studies done in the region identifying synoptic patterns and sea breeze circulations.

**Major Comment 3**: Under high precipitable water conditions, radar signal attenuation may impact cloud and precipitation detection, introducing uncertainties not discussed in the study. For instance, ARSCL faces limitations such as insect contamination and biased cloud-top heights, which could affect the reliability of the results.

This is a great point. In the revised version of the manuscript, we do not use ARSCL to inform the reflectivity threshold we chose. We only use it now to do the quick visual comparison between GOES, KHGX, and ARSCL in Figure 2 and to quantify the cloud mask's skill. In that analysis, we are only using the periods when the lidar detected a cloud base, and we are not using cloud top heights from ARSCL in any way. With regards to the S-band wavelength of KHGX, we believe any attenuation is negligible.

**Major Comment 4**: The key differences between the ESCAPE and TRACER campaigns are not clearly articulated. Considering their different objectives, it is helpful to explain how differences in shallow cloud backgrounds, formation processes, and meteorology between the two campaigns may influence the results for better contextualization.

We added details about the goals of the campaigns to the introduction. ESCAPE focused on convective cloud lifecycles in the broader scope of the environment: urban area, sea breeze circulation, aerosol properties, etc. TRACER focused more on aerosols and understanding aerosol impacts on convective cloud properties.
* * *
**MINOR COMMENTS**

**Minor Comment 1**: The analysis relies on a single observational location (AMF1) for meteorological measurements, which may not fully capture the variability across the domain. Briefly addressing the spatial scale of its representativeness would be helpful.

The meteorological variability figure's biggest flaw was the point location aspect, so we removed it. We agree that a result like that could misrepresent conditions across the domain, so we made our HRRR composite figure instead. The figure highlighted spatial and temporal variability in moisture, temperature, and wind at 850 hPa that correlated with our shallow and deep cloud activity.

**Minor Comment 2**: The lack of a sensitivity analysis for the k-means clustering approach may introduce uncertainties in the identified modes.

A sensitivity analysis would involve changing the value for k. We showed that most of the variance can be accounted for with four modes, and we did not want to introduce more complexity to the analysis by adding more modes. The fact that the results from our four modes align nicely with current studies (Wang et al. 2024) shows that k = 4 was adequate.

**Minor Comment 3**: The manuscript would benefit from more discussion on meteorology, sea breeze circulations, and potential urban influences, complementing the statistical results of clouds and precipitation.

We have added a new figure that shows composites of 850 hPa model output in each cluster, and we discuss the wind patterns, moisture, and temperature and its gradients across the domain as a function of time. We also connect our work to the recent studies done in the region identifying synoptic patterns and sea breeze circulations. With respect to urban influences, we did not have the aerosol data to quantify that kind of impact, and it was beyond the scope of our paper, too.

---

## Referee Report (RR1)

**Review of manuscript egusphere-2024-2984**
**Revision Round 2**

**Title:** Shallow cloud variability in Houston, Texas during the ESCAPE and TRACER field experiments

**Authors:** Zackary Mages, Pavlos Kollias, Bernat Puigdomènech Treserras, Paloma Borque, and Mariko Oue

**Summary:**

The authors have made substantial improvements to the manuscript and the revised version is shaping up nicely into to a final product. I have a few suggestions to address minor gaps that can be resolved quickly. I recommend a minor revision. Red text indicates deletions, and blue text indicates additions. The line numbers in my comments below follow the numbering indicated in the author's tracked changes pdf file.

**Major comments:**

1. **Section 2.4 HRRR data:** Unlike other parts of Section 2, this subsection currently lacks the detailed discussion on the relevance and application of the dataset chosen. Please provide additional context and explanation for choosing HRRR data over observed measurements (or a best-estimate product such as the ARM INTERPSONDE) for this analysis. Specifically, the text should clearly articulate the unique value of HRRR data such as its high-resolution sounding profiles and comprehensive meteorological fields that goes beyond what AMF1 observations offer. Furthermore, because the analysis relies on HRRR model soundings rather than direct observations, it is important to address whether a comparison has been made between HRRR profiles and AMF1 observations. In doing so, please clarify:
   (i)   Which thermodynamic or kinematic features are well represented by the HRRR model.
   (ii)  Which features exhibit significant biases.
   (iii) Whether these biases in the simulated meteorological variables might affect the accuracy of the results.

2. Please be consistent in usage of 'Figs' or 'Figures' throughout the manuscript.

**Minor comments:**

1. **Line 149-150:** …provide a large statistical sample of precipitating and nonprecipitating clouds  over the Houston area…

2. **Line 151:** Daily statistics are used to infer the different spatiotemporal patterns of both shallow and deep convective clouds…

3. **Line 152:** surface properties may not be the most accurate descriptor here. Suggesting replacing with 'surface type' instead.

4. **Line 154-155:** Four main diurnal characteristics  of shallow  clouds will be evaluated over land and water: domain fraction, frequency of occurrence, cloud top height, and cloud to-precipitation ratio.

5. Suggest replacing 'domain fraction' with 'diurnal cloud fraction over land and water' and 'cloud-to-precipitation ratio' with 'precipitation fraction.'

6. **Line 166:** The ARM handbook cited here (Bartholomew, 2020) may be less relevant and perhaps outdated. The latest version of the Laser Disdrometer (LDIS) instrument handbook is likely more useful as a reference. If the authors agree with this assessment, I recommend citing Wand and Bartholomew (2023) instead.

   References:
   Wang, D, and MJ Bartholomew. 2023. Laser Disdrometer (LDIS) Instrument Handbook. U.S. Department of Energy, Atmospheric Radiation Measurement user facility, Richland, Washington. DOE/SCARM-TR-137.

   Handbook available at https://www.arm.gov/capabilities/instruments/ldis

7. **Line 166:** Did you mean to say (warm and cold cloud phase) instead of (warm and cold season)?

8. **Line 167:** …were performed at the AMF1 site.

9. **Line 168:** …followed a  schedule of four radiosonde launches per day at six-hour intervals…

10. **Line 200:** …resolution of  30 m and a …

11. **Line 250:** and  a temporal resolution of 5 minutes

12. **Line 252:** and  a temporal resolution of 5 minutes

13. **Line 269:** …Herbie  Python library (Blaylock, 2024), ...

14. **Line 270:** …wind direction from model analysis (forecast hour 0) here.

15. **Lines 385-386:** The methodology in line 304 states that the KHGX analysis was confined to the region within the 112.5 km radial range (or 225 km diameter). How is the farthest range ~ 176 km then?

16. **Line 402-403:** highest echo-top height corresponding to what reflectivity threshold (0, 10, 20, 30, 40-dBZ)?

**17. Line 767:** Please specify the 'high spatial resolution' explicitly.

**18. Line 768:** Suggest replacing 'forecast hour zero' by 'analysis.'

**19. Lines 777-779:** The description of the temperature gradient orientation is inaccurate. The 850 hPa isotherms in Clusters 1 and 4 are parallel to the coast, thereby making the gradient perpendicular to the coast. Conversely, clusters 2 and 3 have isotherms perpendicular to the coast and thereby the temperature gradient is parallel to the coast. This description also makes logical sense since the sea breeze is expected under a land-ocean temperature contrast forcing perpendicular to the coast (not parallel as suggested in the text). Please revise the text accordingly.

**20. Line 842:** Aided by the onshore flow  shown in Figure 12, we hypothesize…

**21. Lines 851-853:** The first sentence claims that higher shallow cloudiness at 15 LT is found east of HGB area, while the second sentence states the exact opposite. Please clarify and/or rephrase the second sentence.

**22. Line 906-907:**  ($< 0.1$ mm hr$^{-1}$)

**23. Line 909:** …we identify shallow and deep clouds, and associated precipitation

**24. Lines 955-956:** The meteorological set-up we showed in the HRRR composite maps suggest environmental conditions favorable for sea breeze  formation, though to confirm the passage of sea breeze frontal boundaries over the AMF1 site, low-level thermodynamics would be needed

**25. Line 1121:** Suggest either replacing 'data' with 'dataset' or 'is' with 'are'

**Figures:**

**Comment on Figure 4:** The reflectivity data in panel plots (f) and (g) shows some convective cells along the coast which should have been masked out for analysis (marked in red below). Can the authors clarify what's going on here?

---

## Author Response (AR2)

Review of manuscript egusphere-2024-2984 Revision Round 2

Title: Shallow cloud variability in Houston, Texas during the ESCAPE and TRACER field experiments

Authors: Zackary Mages, Pavlos Kollias, Bernat Puigdomènech Treserras, Paloma Borque, and Mariko Oue

Summary:
The authors have made substantial improvements to the manuscript and the revised version is shaping up nicely into to a final product. I have a few suggestions to address minor gaps that can be resolved quickly. I recommend a minor revision. Red text indicates deletions, and blue text indicates additions. The line numbers in my comments below follow the numbering indicated in the author's tracked changes pdf file.

Thank you to Reviewer #1 for their multiple in-depth reviews of our manuscript. The improvements we have made in response have increased the manuscript's quality. We have included all our responses to individual points below.
* * *
**MAJOR COMMENTS**

**Major Comment 1**: (Lines 172-180) Unlike other parts of Section 2, this subsection currently lacks the detailed discussion on the relevance and application of the dataset chosen. Please provide additional context and explanation for choosing HRRR data over observed measurements (or a best-estimate product such as the ARM INTERPSONDE) for this analysis. Specifically, the text should clearly articulate the unique value of HRRR data such as its high-resolution sounding profiles and comprehensive meteorological fields that goes beyond what AMF1 observations offer. Furthermore, because the analysis relies on HRRR model soundings rather than direct observations, it is important to address whether a comparison has been made between HRRR profiles and AMF1 observations. In doing so, please clarify:

    (i)  Which thermodynamic or kinematic features are well represented by the HRRR model.
    (ii)  Which features exhibit significant biases.
    (iii)  Whether these biases in the simulated meteorological variables might affect the accuracy of the results.

Thank you for this comment. To preserve the paper's flow, we decided to put our response to this comment in Section 4.5 rather than Section 2.4. In Section 4.5, before we discuss the results, we provide some justification as to why we are using the HRRR instead of the instrumentation at the AMF1. We made the following addition: "We use the meteorological observations provided by the soundings launched from the AMF1 site to establish our shallow vs deep classification in the radar data. The soundings and other instrumentation at the AMF1 site would be useful to quantify meteorological variability in the clusters, but point observations would limit our ability to look at variability across our large domain. So, we use HRRR model output, which has high spatial resolution of 3 km, to provide this meteorological context as the best alternative to real

observations. We compare the HRRR soundings at the grid point closest to the AMF1 site to the real soundings and find the median correlation coefficients for temperature, moisture, and wind speed to be high (>0.8; not shown). Even with this strong performance, we only qualitatively interpret the spatial patterns shown in the HRRR model data in our subsequent analysis."

Here is the correlation analysis we mention if you are curious. We perform nearest neighbor interpolation to map the real observations onto the model's vertical grid. Sample sizes are in parentheses.

**Median correlation coefficients between TRACER soundings and the nearest HRRR grid point to the sites at the launch hour**

| | 0 LT | 6 LT | 11 LT | 12 LT | 13 LT | 14 LT | 15 LT | 16 LT | 17 LT | 18 LT |
|---|---|---|---|---|---|---|---|---|---|---|
| M1 Temperature | 0.9997 (121) | 0.9998 (122) | 0.9998 (2) | 0.9997 (120) | 0.9997 (2) | 0.9998 (38) | 0.9998 (39) | 0.9998 (3) | 0.9998 (36) | 0.9998 (116) |
| S3 Temperature | | | | 0.9997 (39) | 0.9992 (1) | 0.9998 (36) | 0.9997 (38) | 0.9996 (1) | 0.9997 (38) | 0.9997 (36) |
| M1 Specific Humidity | 0.9857 (121) | 0.9878 (122) | 0.9824 (2) | 0.9872 (120) | 0.9936 (2) | 0.9844 (38) | 0.9877 (39) | 0.9808 (3) | 0.9886 (36) | 0.9867 (116) |
| S3 Specific Humidity | | | | 0.9831 (39) | 0.9903 (1) | 0.9873 (36) | 0.9825 (38) | 0.9892 (1) | 0.9886 (38) | 0.9882 (36) |
| M1 Wind Speed | 0.8922 (121) | 0.9074 (122) | 0.8422 (2) | 0.8877 (120) | 0.8815 (2) | 0.8307 (38) | 0.8536 (39) | 0.8722 (3) | 0.8689 (36) | 0.8862 (116) |
| S3 Wind Speed | | | | 0.8556 (39) | 0.7538 (1) | 0.8472 (36) | 0.7888 (38) | 0.5365 (1) | 0.8818 (38) | 0.8406 (36) |

M1 = Main Site
S3 = Ancillary Site

**Major Comment 2**: Please be consistent in usage of 'Figs' or 'Figures' throughout the manuscript.

Thank you for this comment. We changed the way we referenced figures throughout the manuscript. We abbreviated to Fig. when it appeared in running text, and we only stayed with Figure when it began a sentence. When putting the references to figure numbers in parentheses, we abbreviated to a singular "Fig." in all instances where we referenced a singular figure, even if we referenced multiple panels on that figure. We abbreviated to "Figs." when we referenced different figures.
* * *
**MINOR COMMENTS**

**Minor Comment 1**: (Line 149-150) ...provide a large statistical sample of precipitating and nonprecipitating clouds  over the Houston area...

We made this change.

**Minor Comment 2**: (Line 151) Daily statistics are used to infer the different spatiotemporal patterns of both shallow and deep convective clouds...

We made this change.

**Minor Comment 3**: (Line 152) surface properties may not be the most accurate descriptor here. Suggesting replacing with 'surface type' instead.

We changed to "surface type" instead.

**Minor Comment 4**: (Line 154-155) Four main diurnal characteristics  of shallow  clouds will be evaluated over land and water: domain fraction, frequency of occurrence, cloud top height, and cloud to-precipitation ratio. Also suggest replacing 'domain fraction' with 'diurnal cloud fraction over land and water' and 'cloud-to-precipitation ratio' with 'precipitation fraction.'

We changed to "Four main diurnal characteristics of shallow clouds will be evaluated over land and water: cloud fraction, frequency of occurrence, cloud top height, and precipitation occurrence."

**Minor Comment 5**: (Line 166) The ARM handbook cited here (Bartholomew, 2020) may be less relevant and perhaps outdated. The latest version of the Laser Disdrometer (LDIS) instrument handbook is likely more useful as a reference. If the authors agree with this assessment, I recommend citing Wang and Bartholomew (2023) instead.
    References:
    Wang, D, and MJ Bartholomew. 2023. Laser Disdrometer (LDIS) Instrument Handbook. U.S. Department of Energy, Atmospheric Radiation Measurement user facility, Richland, Washington. DOE/SCARM-TR-137.
    Handbook available at https://www.arm.gov/capabilities/instruments/ldis

We agree. We will update the citation. The handbook listed with the dataset on the ARM Archive has changed since we started putting this manuscript together. Thank you for catching this!

**Minor Comment 6**: (Line 166) Did you mean to say (warm and cold cloud phase) instead of (warm and cold season)?

We meant "(liquid and frozen)". We made this change.

**Minor Comment 7**: (Line 167) ...were performed at the AMF1 site.

We made this change.

**Minor Comment 8**: (Line 168) ...followed a  schedule of four radiosonde launches per day at six-hour intervals...

We made this change.

**Minor Comment 9**: (Line 200) ...resolution of  30 m and a ...

We made this change.

**Minor Comment 10**: (Line 250) and  a temporal resolution of 5 minutes

We made this change.

**Minor Comment 11**: (Line 252) and  a temporal resolution of 5 minutes

We made this change.

**Minor Comment 12**: (Line 269) ...Herbie  Python library (Blaylock, 2024), ...

We made this change.

**Minor Comment 13**: (Line 270) ...wind direction from model analysis (forecast hour 0) here.

We changed to "wind direction from model analysis at forecast hour 0 here".

**Minor Comment 14**: (Lines 385-386) The methodology in line 304 states that the KHGX analysis was confined to the region within the 112.5 km radial range (or 225 km diameter). How is the farthest range ~ 176 km then?

Thank you for catching this. We clarified the statement to read "At the furthest range considered in this study (112.5 km from KHGX), the minimum detectable signal is between 0 and 5 dBZ, so our threshold is applicable."

**Minor Comment 15**: (Line 402-403) highest echo-top height corresponding to what reflectivity threshold (0, 10, 20, 30, 40-dBZ)?

We are not using a reflectivity threshold. We only identify the height of the last reflectivity echo, especially because data becomes sparse at those higher altitudes and the exact height of a reflectivity threshold would have to be interpolated somehow. We clarified: "comparing the highest KHGX echo top height (no reflectivity threshold considered) at that pixel…"

**Minor Comment 16**: (Line 767) Please specify the 'high spatial resolution' explicitly.

We changed to "Due to its high spatial resolution of 3 km".

**Minor Comment 17**: (Line 768) Suggest replacing 'forecast hour zero' by 'analysis.'

We made this change.

**Minor Comment 18**: (Lines 777-779) The description of the temperature gradient orientation is inaccurate. The 850 hPa isotherms in Clusters 1 and 4 are parallel to the coast, thereby making the gradient perpendicular to the coast. Conversely, clusters 2 and 3 have isotherms perpendicular to the coast and thereby the temperature gradient is parallel to the coast. This description also makes logical sense since the sea breeze is expected under a land- ocean temperature contrast forcing perpendicular to the coast (not parallel as suggested in the text). Please revise the text accordingly.

Thank you for noting this. We were incorrect in our conventions. We changed the text to read: "The isotherms are oriented parallel to the coast all day in C1 and C4 to create a temperature gradient perpendicular to the coast, which, combined with the winds and moisture, favors sea breeze conditions. Despite a parallel-to-the-coast temperature gradient orientation at 9 LT in C2 and C3 (Figure 12b-c), the gradient attains a more perpendicular-to-the-coast orientation later in the day, although it is not as pronounced as the ones shown in C1 and C4."

**Minor Comment 19**: (Line 842) Aided by the onshore flow  shown in Figure 12, we hypothesize...

We made this change.

**Minor Comment 20**: (Lines 851-853) The first sentence claims that higher shallow cloudiness at 15 LT is found east of HGB area, while the second sentence states the exact opposite. Please clarify and/or rephrase the second sentence.

Thank you for catching this; it was not clearly worded. We changed to: "In C4, two noticeable spatial patterns are observed at 15 and 17 LT (Fig. 13j-k), and these patterns are not associated with the coastline and the surface type transition. First, over land, higher shallow cloud frequencies (24-40%) are observed east of HGB while, west of the HGB area, lower shallow cloud frequencies (10-25%) are observed (Fig 13j-k). Second, at 15 LT, the maximum shallow cloudiness is observed over land (Fig. 13j), and, at 17 LT as well as at 19 LT, the maximum shallow cloudiness frequency is observed over water unlike in C1 (>40%; Fig. 13k-l)."

**Minor Comment 21**: (Line 906-907)  (< 0.1 mm hr-1)

We made this change.

**Minor Comment 22**: (Line 909) ...we identify shallow and deep clouds, and associated precipitation

We made this change.

**Minor Comment 23**: (Lines 955-956) The meteorological set-up we showed in the HRRR composite maps suggest environmental conditions favorable for sea breeze  formation, though to confirm the passage of sea breeze frontal boundaries over the AMF1 site, low-level thermodynamics would be needed

We made this change.

**Minor Comment 24**: (Line 1121) Suggest either replacing 'data' with 'dataset' or 'is' with 'are'

We made this change: "The HRRR dataset is acquired using…"

**Minor Comment 25**: (Figure 4) The reflectivity data in panel plots (f) and (g) shows some convective cells along the coast which should have been masked out for analysis (marked in red below). Can the authors clarify what's going on here?

Several of the cells seen in Fig. 4e are not included in either Fig. 4f or Fig. 4g because they fall outside the range ring of our study. The low reflectivity, more stratiform edges of these clouds are not visible in Fig. 4f,g either, so we are focusing on the precipitation cores themselves. Beyond this, every precipitation core we see in Fig. 4e is accounted for in the cumulative view of Fig. 4f and 4g. We often see multicellular precipitation structures in the radar data, which accounts for why parts of the structures are classified as shallow and as deep and get split up.